# Methane emissions offset atmospheric carbon dioxide uptake in coastal macroalgae, mixed vegetation and sediment ecosystems

Florian Roth [1,2] ✉, Elias Broman [1,3], Xiaole Sun [1,4], Stefano Bonaglia [5], Francisco Nascimento [1,3], John Prytherch[6], Volker Brüchert [7,8], Maysoon Lundevall Zara[7], Märta Brunberg[2], Marc C. Geibel [1], Christoph Humborg[1,2] & Alf Norkko[1,2]

Coastal ecosystems can efficiently remove carbon dioxide ($CO_2$) from the atmosphere and are thus promoted for nature-based climate change mitigation. Natural methane ($CH_4$) emissions from these ecosystems may counterbalance atmospheric $CO_2$ uptake. Still, knowledge of mechanisms sustaining such $CH_4$ emissions and their contribution to net radiative forcing remains scarce for globally prevalent macroalgae, mixed vegetation, and surrounding depositional sediment habitats. Here we show that these habitats emit $CH_4$ in the range of 0.1 – 2.9 mg $CH_4$ $m^{-2}$ $d^{-1}$ to the atmosphere, revealing in situ $CH_4$ emissions from macroalgae that were sustained by divergent methanogenic archaea in anoxic microsites. Over an annual cycle, $CO_2$-equivalent $CH_4$ emissions offset 28 and 35% of the carbon sink capacity attributed to atmospheric $CO_2$ uptake in the macroalgae and mixed vegetation habitats, respectively, and augment net $CO_2$ release of unvegetated sediments by 57%. Accounting for $CH_4$ alongside $CO_2$ sea-air fluxes and identifying the mechanisms controlling these emissions is crucial to constrain the potential of coastal ecosystems as net atmospheric carbon sinks and develop informed climate mitigation strategies.

Climate change mitigation demands reduced anthropogenic emissions of greenhouse gases (GHGs) and effective removal of excess carbon dioxide ($CO_2$) and methane ($CH_4$) from the atmosphere. Shallow-water coastal ecosystems can absorb and store large amounts of carbon from the atmosphere through vegetation buildup and long-term sediment burial[1]. This ecosystem function has raised worldwide interest in the scientific community, conservation organizations, and governmental bodies about the potential of these ecosystems in short-term climate mitigation[2–4]. In fact, restoring the carbon sequestration capacity of coastal ecosystems and improving their global management could result in an annual uptake of 841 (621–1064) Tg $CO_2$-equivalents ($CO_2$-eq.) per year, representing a significant GHG sink in the global carbon budget[4]. However, some of this organic carbon is metabolized and returned to the atmosphere as $CH_4$[5,6]. While aquatic $CH_4$ emissions can partly offset the GHG sink estimate of the terrestrial landscape[7] and of some vegetated coastal ecosystems, such as mangroves[8], the magnitude of $CH_4$ fluxes and their contribution to the net atmospheric GHG exchange remains unknown for the majority of

[1]Baltic Sea Centre, Stockholm University, Stockholm, Sweden. [2]Tvärminne Zoological Station, Faculty of Biological and Environmental Sciences, University of Helsinki, Hanko, Finland. [3]Department of Ecology, Environment and Plant Sciences, Stockholm University, Stockholm, Sweden. [4]Center of Deep Sea Research, Institute of Oceanology, Chinese Academy of Sciences, Qingdao, China. [5]Department of Marine Sciences, University of Gothenburg, Gothenburg, Sweden. [6]Department of Meteorology, Stockholm University, Stockholm, Sweden. [7]Department of Geological Sciences, Stockholm University, Stockholm, Sweden. [8]Bolin Centre for Climate Research, Stockholm University, Stockholm, Sweden. ✉e-mail: florian.roth@su.se

coastal environments and challenges our ability to develop informed climate mitigation strategies for these ecosystems[9].

Primary production from submerged and partially emerged coastal vegetation[10–12] and the associated accumulation of allochthonous carbon within or outside of these systems makes coastal environments some of the most carbon-rich ecosystems in the world[13–15]. While the high net ecosystem productivity is a main driver for the uptake of atmospheric $CO_2$ in vegetated ecosystems[16], the direct contribution to the sea-air $CO_2$ exchange remains understudied across many coastal environments[17]. In addition, methanogenic archaea in anoxic sediments associated with these ecosystems produce $CH_4$ during carbon burial, which adds to the release of locally fixed autochthonous and imported allochthonous carbon from these ecosystems[18,19]. To date, it is estimated that the coastal ocean contributes 5 – 28 Tg $CH_4$ yr$^{-1}$ to total global $CH_4$ emissions[6,20]. Because $CO_2$ and $CH_4$ differ in their atmospheric lifetimes and radiative efficiencies – with $CH_4$ having a sustained-flux global warming potential (SGWP) 45 times more potent than $CO_2$ over a 100-year time horizon[21] – $CH_4$ emissions may substantially counterbalance the carbon sink capacity attributed to the local instantaneous atmospheric $CO_2$ uptake[22].

Yet, the majority of coastal ecosystem carbon assessments are still based on changes in sediment and biomass carbon inventories[23] or primary productivity measurements using oxygen[24] to infer GHG exchange with the atmosphere. These conventional methods assume that emissions to the atmosphere are in the form of $CO_2$, neglecting the part of the carbon pool that is metabolized to $CH_4$ with a higher radiative forcing. In addition, neither carbon burial nor benthic productivity necessarily leads to a direct net uptake of atmospheric $CO_2$ over the same surface area for which the carbon burial is considered. This is because, first, the water column separates the atmosphere from benthic systems and sea-air $CO_2$ gas exchange is also affected by lateral inorganic carbon inputs, buffer effects, and the residence time of $CO_2$ in the water column[16,25]. Second, buried sedimentary organic carbon is composed both of laterally imported allochthonous and locally produced autochthonous sources[26]. This implies that even if a system is net autotrophic and exhibits carbon burial, it may still function as a net source of atmospheric $CO_2$ locally if the remineralization of laterally imported carbon is high.

Simultaneous and continuous $CO_2$ and $CH_4$ sea–air flux measurements are, therefore, indispensable to determine whether a coastal ecosystem acts as a net source or sink of atmospheric carbon-based GHGs – that is, if it has a positive or negative effect on radiative forcing[21]. In situ automated cavity ring-down spectroscopy is particularly effective to quantify coastal sea-air $CO_2$ and $CH_4$ fluxes simultaneously – but its application has been limited to estuarine, mangrove, and seagrass systems[27–30]. The paucity of similar measurements across a wider range of coastal environments – for example, globally prevalent and highly productive macroalgae and mixed vegetation habitats, or their surrounding shallow depositional sediment areas – currently complicates efforts evaluating the realized potential of our coasts to remove carbon from the atmosphere. This is because (1) rigorous evidence for the uptake of atmospheric $CO_2$ by many such coastal systems through direct sea-air $CO_2$ gas exchange remains understudied[17], and (2) concurrent $CH_4$ emissions from these environments could offset or even negate their value as atmospheric carbon dioxide sinks[7].

In this study, we quantify hourly, daily, and seasonal sea-air $CO_2$ and $CH_4$ fluxes simultaneously using a fast-response automated gas equilibrator system for in situ continuous measurements across three globally prevalent shallow water (<4 m) coastal ecosystems to assess their direct contribution to sea-air GHG exchange. To represent systems with varying carbon dynamics, which are also underrepresented with regards to simultaneous and continuous $CO_2$ and $CH_4$ sea-air flux measurements, we quantify GHG fluxes in a northern temperate area of

the Baltic Sea in the following habitats: (a) 'macroalgae' (i.e., mainly *Fucus vesiculosus*) areas, which have a high primary production (and carbon sequestration) potential[31–34] but are usually not associated with $CH_4$ emissions due to the lack of sediments on rocky substrates; (b) areas with submerged 'mixed vegetation' (i.e., a mix of macrophytes and macroalgae) on soft sediments, which trap large amounts of allochthonous and autochthonous carbon; and (c) adjacent 'bare sediments' with marginal vegetation (<10% total vegetation, of which were mainly dislodged *F. vesiculosus*) that are common deposition sites with low primary productivity but a high potential of carbon remineralization. We complement in situ GHG measurements with sediment geochemistry and microbial community structure and diversity (16S rRNA gene sequencing) assessments to describe site-specific mechanism by which $CH_4$ emissions are sustained in these habitats.

## Results and discussion

### Dynamic $CH_4$ and $CO_2$ sea-air gas exchange across coastal ecosystems

High-resolution sea-air $CH_4$ flux measurements revealed that all habitats were net sources of $CH_4$ to the atmosphere during all study periods. There was high variability in the magnitude of these fluxes across habitats and seasons (Fig. 1a and Supplementary Table 1; statistics in Supplementary Table 2). We report in situ $CH_4$ sea-air fluxes in the order of $0.1 \pm 0.0$ to $1.8 \pm 0.1$ mg $CH_4$ m$^{-2}$ d$^{-1}$ (mean ± SE) from the macroalgae habitat. $CH_4$ emissions from macroalgae have previously only been reported from in vitro studies where macroalgae material was artificially fermented[35,36] or in situ from natural degradation of macroalgae (i.e., *Fucus vesiculosus*) on beaches as beach wrack[37]. Daily mean net $CH_4$ fluxes in the mixed vegetation and bare sediment areas ranged from $0.1 \pm 0.0$ to $2.9 \pm 0.3$ mg $CH_4$ m$^{-2}$ d$^{-1}$ and $0.1 \pm 0.0$ to $2.5 \pm 0.2$ mg $CH_4$ m$^{-2}$ d$^{-1}$, respectively (Supplementary Table 1). The magnitude of these fluxes corroborates recent measurements in similar coastal environments with fluxes ranging from 0.6 to 8.3 mg $CH_4$ m$^{-2}$ d$^{-1}$ (ref. 38). Across all habitats, $CH_4$ fluxes were one order of magnitude higher in summer and fall than in spring and winter (Supplementary Table 1). As methanogenesis exceeds $CH_4$ oxidation during warm periods[39,40], increased $CH_4$ production in summer and fall likely explains this observation. $CH_4$ emissions in summer and fall are also comparable to globally compiled values for other coastal vegetated systems, such as mangroves, salt marshes, and seagrass beds (i.e., median, range: 4.5, −1.1–1168.8; 3.6, −1.5–1509.8; and 1.0, 0.0–6.4 mg $CH_4$ m$^{-2}$ d$^{-1}$, respectively)[5]. Over an annual cycle, the cumulative net fluxes of $CH_4$ to the atmosphere were 0.34 (±0.01) g $CH_4$ m$^{-2}$ y$^{-1}$ in the macroalgae, 0.55 (±0.03) g $CH_4$ m$^{-2}$ y$^{-1}$ in the mixed vegetation, and 0.38 (±0.02) g $CH_4$ m$^{-2}$ y$^{-1}$ in the surrounding bare sediment areas (data presented as cumulative annual net flux and propagated error using daily means and the associated uncertainty). Such spatial and temporal heterogeneity of $CH_4$ flux distribution in coastal ecosystems suggest that high-resolution measurements are urgently needed to improve the reliability of $CH_4$ estimates and confine the habitat-specific contribution to regional and global $CH_4$ budgets[40].

Daily mean net sea-air $CO_2$ fluxes ranged from $-763 \pm 99$ mg $CO_2$ m$^{-2}$ d$^{-1}$ (mean ± SE; sink of atmospheric $CO_2$) to $390 \pm 35$ mg $CO_2$ m$^{-2}$ d$^{-1}$ (source of atmospheric $CO_2$), with significant differences across habitats and seasons (Supplementary Table 1; statistics in Supplementary Table 3). The macroalgae and mixed vegetation sites were net sinks of atmospheric $CO_2$ over diel cycles in spring, summer, and fall (Supplementary Table 1). The hourly fluxes displayed considerable temporal dynamics over diel cycles, with peak $CO_2$ uptake rates usually measured between 13:00–17:00 h (Fig. 1b) in the vegetated areas. The data confirms that sunlight stimulated photosynthetic activity of submerged vegetation, causing an undersaturation of $pCO_2$ in surface waters relative to the atmospheric equilibrium, and promoted direct

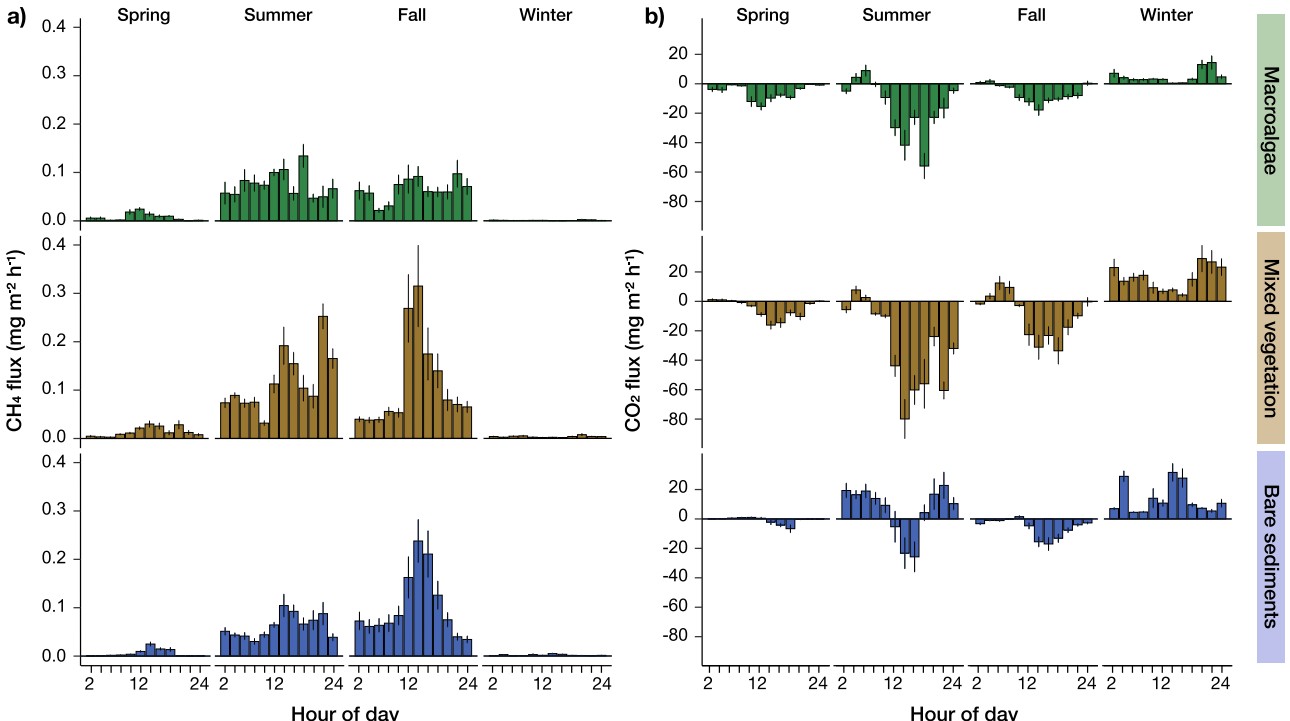

**Fig. 1 | Hourly methane and carbon dioxide sea-air fluxes in coastal macroalgae, mixed vegetation, and bare sediment habitats.** Hourly sea-air $CH_4$ (**a**) and $CO_2$ (**b**) fluxes across four seasons in three coastal ecosystems. Values are means ± standard error. Positive fluxes refer to an efflux from the water to the atmosphere (source), while negative fluxes depict an uptake of atmospheric GHGs (sink). In situ continuous (1 Hz) measurements were averaged to 15 min intervals and binned in 2-hour blocks for graphical representation. Daily integrated net sea-air fluxes of $CH_4$ and $CO_2$ across seasons and habitats are presented in the text and Supplementary Table 1.

net $CO_2$ removal from the atmosphere[16,41,42]. The surrounding bare sediment areas were moderate sinks of $CO_2$ in spring and fall ($-21 \pm 8$ and $-132 \pm 17$ mg $CO_2$ m$^{-2}$ d$^{-1}$, respectively) compared to the vegetated habitats, and they became net sources of $CO_2$ in summer ($157 \pm 59$ mg $CO_2$ m$^{-2}$ d$^{-1}$). In photic sediments, ecosystem respiration can quickly outbalance or even outweigh $CO_2$ removal via photosynthesis, as photosynthetic activity remains limited to low-biomass microphytobenthos or dislodged macrophytes[32]. All habitats became net sources of atmospheric $CO_2$ over diel cycles in winter (Fig. 1b and Supplementary Table 1), suggesting a dominance of heterotrophic over autotrophic processes through suppressed photosynthetic activity in cold waters and insufficient light availability during this period of the year[32,43]. Over an annual cycle, the macroalgae and mixed vegetation habitats acted as net sinks of atmospheric $CO_2$ with cumulative fluxes of $-52$ ($\pm 5$) and $-71$ ($\pm 10$) g $CO_2$ m$^{-2}$ y$^{-1}$, respectively (data presented as cumulative annual net flux and propagated error using daily means and the associated uncertainty). In comparison, the bare sediments were net sources of $CO_2$ to the atmosphere with 30 ($\pm 6$) g $CO_2$ m$^{-2}$ y$^{-1}$. The data confirm high rates of primary production across macrophyte and macroalgae canopies in similar geographic regions[31,32,44] and highlight that net ecosystem production in macroalgae beds and other submerged mixed vegetation can directly translate into atmospheric $CO_2$ removal[16,42,45].

## $CH_4$ emissions offset the carbon sink capacity attributed to atmospheric $CO_2$ uptake

The simultaneous measurement approach of $CH_4$ and $CO_2$ sea-air fluxes is viable for directly comparing the direction and magnitude of carbon-based sea-air gas exchange and better constraining the net radiative balance of coastal habitats. We calculated the net sea-air GHG balance by converting $CH_4$ fluxes into $CO_2$-eq. fluxes based on the SGWP over a 100-year time horizon[21]. This measure describes metric-weighted GHG exchanges[22], i.e., the net $CO_2$-eq. flux is the sum of the $CO_2$-eq. fluxes of each gas (i.e., $CO_2$ and $CH_4$). The SGWP was chosen

since these coastal ecosystems continually exchange GHGs with the atmosphere, which is not captured by the one-time "pulse" emission basis of the global warming potential (GWP)[21]. In the following, the carbon sink capacity attributed to atmospheric $CO_2$ uptake refers to an instantaneous influx of $CO_2$ from the atmosphere into the water caused by undersaturation of $pCO_2$ in surface waters relative to the atmospheric equilibrium; note, this capacity does not relate to long-term carbon sequestration processes as burial or export.

We found that $CO_2$-eq. $CH_4$ fluxes substantially offset the carbon sink capacity attributed to the net atmospheric $CO_2$ uptake (Fig. 2a, b); however, the magnitude of this offset was variable across habitat types and seasons, depending on the magnitude of $CO_2$ relative to $CO_2$-eq. $CH_4$ fluxes (Fig. 2a). For example, some of the highest offset (i.e., 84%; Fig. 2 and Supplementary Table 1) was observed in the bare sediment habitats, where photosynthetic activity by microphytobenthos or dislodged macrophytes (leading to $CO_2$ uptake) is counterbalanced by ecosystem respiration (leading to $CO_2$ release)[32], and $CH_4$ fluxes are sustained by organic matter-rich soft sediments[46]. In contrast, highly productive macroalgae (leading to increased rates of $CO_2$ uptake) and marginal $CH_4$ emissions showed generally lower offsets in the carbon sink capacity attributed to atmospheric $CO_2$ uptake by concurrent $CH_4$ emissions. Seasonally, the greatest offsets by $CO_2$-eq. $CH_4$ fluxes were observed in summer and fall, reducing the net GHG balance by 21–44% in the macroalgae, 16–47% in the mixed vegetation, and 42–84% in the bare sediment habitats. In general, the offset was higher in fall due to the seasonal asynchronicity between $CO_2$ and $CH_4$ inventories (Fig. 2), which highlights the limitations of simple empirical functions, such as temperature relationships, for predicting ecosystem GHG fluxes[47]. Across all habitats, the contribution of $CO_2$-eq. $CH_4$ fluxes to the net GHG balance were marginal in winter (~1%), likely due to low $CH_4$ production at low temperatures[39]. Over an annual cycle, $CO_2$-eq. $CH_4$ fluxes lowered the net atmospheric $CO_2$-eq. sink capacity attributed to $CO_2$ uptake in the macroalgae habitats from $-52$ down to $-38$ g $CO_2$-eq. m$^{-2}$ y$^{-1}$ (i.e., 28% reduction) and from $-71$ down to $-46$ g $CO_2$-eq. m$^{-2}$

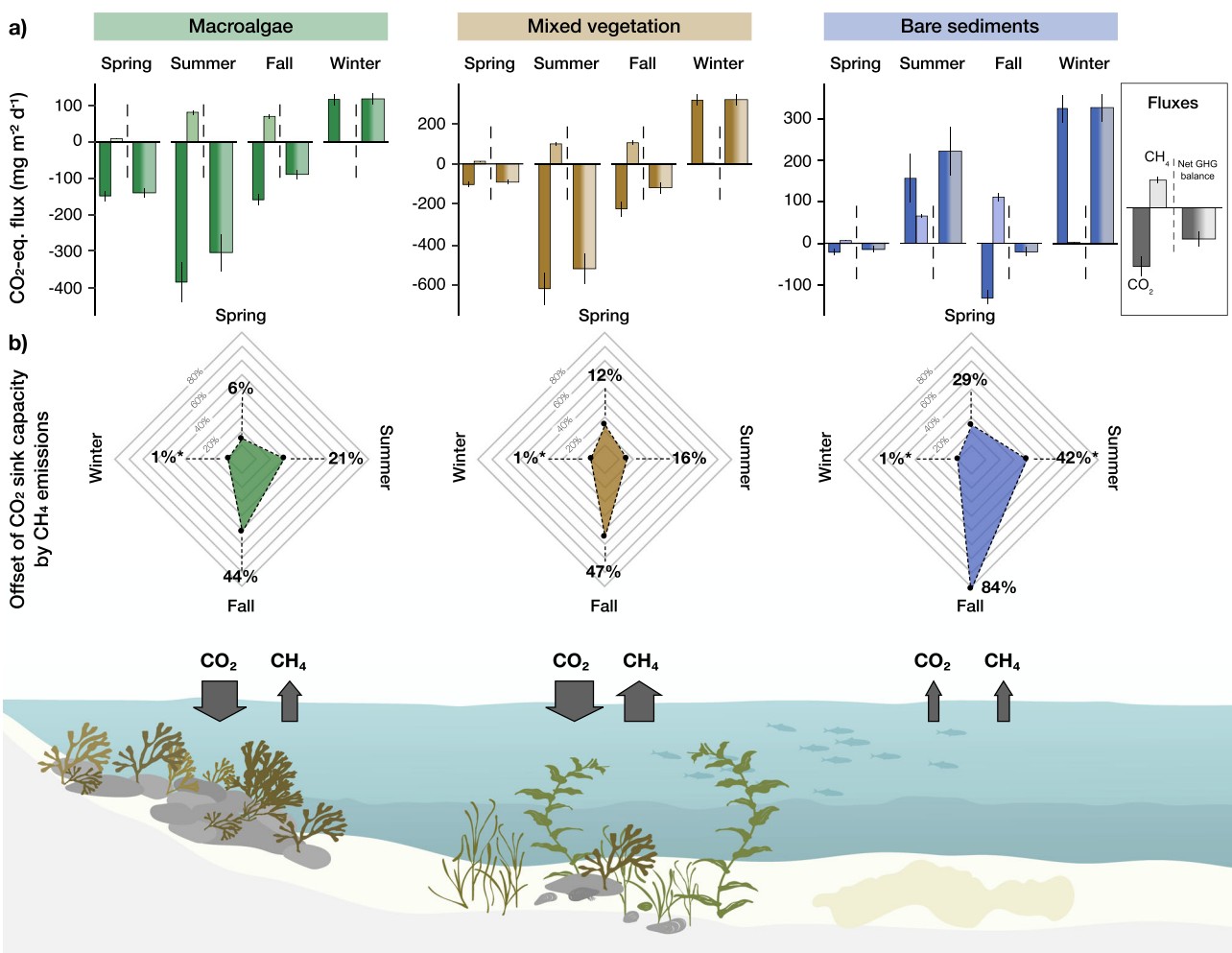

**Fig. 2 | Seasonal net greenhouse gas balances.** Daily mean net fluxes of $CO_2$, $CH_4$, and the net greenhouse gas balance (all expressed in $CO_2$-eq. fluxes) (**a**), and the offset (in %) of the carbon sink capacity attributed to atmospheric $CO_2$ uptake by $CO_2$-eq $CH_4$ emissions (**b**). Values in **a** are means ± standard error. Positive fluxes refer to an efflux from the water to the atmosphere (source), while negative fluxes depict an uptake of atmospheric GHGs (sink). $CO_2$-equivalent $CH_4$ fluxes were calculated using the sustained-flux global warming potential (SGWP) on a 100-year time horizon of 45[21]. The net greenhouse gas balance is calculated based on net $CO_2$ and net $CO_2$-eq. $CH_4$ fluxes. The offset in (b) is calculated from the net $CO_2$ flux and the net $CO_2$-eq. $CH_4$ flux. An offset implies that the carbon sink capacity attributed to atmospheric $CO_2$ uptake was counterbalanced by concomitant $CO_2$-eq. $CH_4$ fluxes; an offset denoted with asterisk (*) implies that $CO_2$ emissions to the atmosphere were increased by the $CO_2$-eq. $CH_4$ flux. Note the different scales on the y-axis. Arrows at the water interface are conceptual and depict prevailing flux direction and magnitude in each habitat. Data for the daily net sea-air fluxes of $CO_2$, $CH_4$, $CO_2$-eq. of $CH_4$, and the net greenhouse gas (GHG) across four seasons in three coastal ecosystems are presented in Supplementary Table 1. Abbreviations: GHG = greenhouse gas. Original artwork by Elsa Wikander at Azote AB.

$y^{-1}$ (i.e., 35% reduction) in the mixed vegetation habitats. Cumulative $CO_2$-eq. $CH_4$ fluxes augmented the positive net sea-air $CO_2$ fluxes of bare sediments by 57% (i.e., from 30 to 47 g $CO_2$-eq. $m^{-2}$ $y^{-1}$).

Offsets in the coastal carbon sink capacity attributed to atmospheric $CO_2$ uptake by concurrent $CH_4$ emissions remain uncertain and difficult to compare because of the limited assessments in a few coastal ecosystems, such as mangroves[8] and seagrasses[48–50]. Our results show that $CH_4$ emissions from globally prevalent coastal habitats with unvegetated sediments, but also with productive macroalgae and mixed vegetation can lower the GHG sink estimate attributed to the atmospheric $CO_2$ uptake by one-third over an annual cycle. Thus, accounting for $CH_4$ alongside $CO_2$ sea-air fluxes becomes indispensable to correctly quantify the potential of coastal ecosystems to act as net atmospheric carbon sinks, which is necessary to develop informed climate mitigation strategies.

### Distinct microbial communities shape habitat-specific $CH_4$ dynamics

High rates of $CH_4$ emissions have been ascribed to habitats with mixed vegetation[38,51] and surrounding depositional[52,53] areas with organic matter-rich soft sediments. In general, these and similar coastal sediment systems account for the majority of total marine $CH_4$ emissions[20]. Rapid organic matter and sediment accumulation rates, deep anoxic sediment layers, bottom water oxygen depletion, and shallow sulfate-methane transition zones acting as "$CH_4$-filter" can all contribute to increased $CH_4$ release rates from coastal sediments[15,46,54]. The high rates of $CH_4$ emissions from the macroalgae habitats in our study are therefore intriguing because of the prevalence on rocky hard-bottom substrates and the absence of the above-mentioned "classical" sedimentary conditions that promote $CH_4$ formation. To examine whether the high surface water $CH_4$ concentrations (Supplementary Table 4) in the macroalgae habitat were a consequence of lateral transport of dissolved $CH_4$ from neighboring habitats or a unique feature of this particular location, we performed three additional assessments:

First, we sampled bottom substrates of all habitats across seasons to evaluate the local geochemical and microbial potential for $CH_4$ production. We found organic matter-rich anoxic sediments, with similar organic carbon contents (3–5%) below 10 cm depth in all habitats and months (Supplementary Fig. 1; selected results from summer are shown in Fig. 3a). While macroalgae grew on rocks,

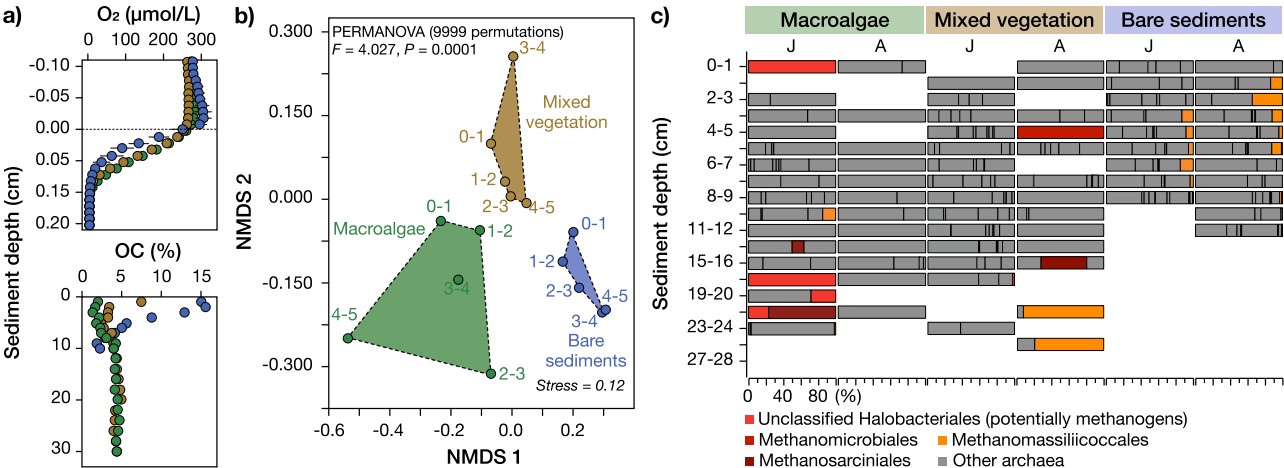

**Fig. 3 | Habitat-specific sediment biogeochemical and microbial characteristics.** Sediment biogeochemical (**a**) and microbial (**b**, **c**) characteristics during the sampling in summer. (**a**) Sediment microprofiles for $O_2$ and organic carbon (OC) contents. (**b**) NMDS plot of the Bray-Curtis dissimilarity index between the three habitats. The results are based on the whole 16S rRNA gene identified microbial community. The plot shows data from the first 5 cm layers in the sediment. The *pseudo-F* and *P* values show the statistical results from PERMANOVA (9999 permutations) based on testing all three habitats together. **c** Stacked bars showing the known methanogenic archaeal community (according to the SILVA v138.1 database) on the lowest taxonomic classified level during the sampling in July (J) and August (A). The y-axis shows the different sediment layers, while the x-axes show the relative abundance (% of all archaea, labels only shown on first x-axis). "Other archaea" denote groups <0.1% average of all samples. Taxonomic labels in color denote known methanogens in the dataset according to scientific literature. Note that for the station "bare sediments", samples could only be collected down to 9 cm (July) and 12 cm (August). Empty cells denote that no archaea were detected in the dataset.

organic matter-rich sediment was present in sediment pockets underlying the canopy, between rocks and boulders (exemplary picture in Supplementary Fig. 2a). Local organic carbon-rich deposits with anoxic conditions may be conducive to methanogenesis[13]. The presence of methanogens in these sediments was confirmed with 16S rRNA gene amplicon sequencing. However, the three habitats showed distinctively different microbial community structures in the top 5 cm of the sediment during all four seasons (Bray-Curtis dissimilarity analyses, PERMANOVA, *P* = 0.0001; Supplementary Figs. 3 and 4; example results from summer are shown in Fig. 3b), and methanogenetic archaea were detected at all sites (Fig S5; exemplifying results from summer shown in Fig. 3c). Some of the retrieved 16S rRNA gene sequences in the mixed vegetation and macroalgae systems were affiliated with classical methanogenic orders such as Methanomicrobiales (i.e., using hydrogen to produce $CH_4$) and methylotrophic Methanosarcinales (i.e., using one-carbon compounds such as methanol) (Supplementary Fig. 5; selected results from summer are shown in Fig. 3c)[55]. We also found unclassified sequences within the archaeal phylum Crenarchaeota (including the class Bathyarchaeia) in the macroalgae habitat (Supplementary Fig. 5). Some of these archaea encode the universal phylogenetic marker gene *mcrA* used to identify methanogenic microorganisms[56]. The bare sediments showed a prevalence of methylotrophic methanogens of the order Methanomassiliicoccales near the sediment surface (Supplementary Fig. 5; results from summer shown in Fig. 3b), which are major $CH_4$ producers in mangroves systems[57]. Finally, deeper oxygen penetration depths (Supplementary Fig. 1) in the mixed vegetation habitat are consistent with a higher relative abundance of methane-oxidizing Methylococcales bacteria, suggesting a more efficient sedimentary $CH_4$ oxidation filter in this habitat compared to the macroalgae and bare sediment area (Supplementary Fig. 6). Together, the data show that distinct methanogenic communities in vegetated and unvegetated habitats are likely involved in producing $CH_4$ in sediments or sediment pockets between rocks locally[50].

Second, we took samples of organic matter aggregates (floating filamentous algal and/or organic matter debris) associated with dense stands of macroalgae (Supplementary Fig. 2a, b) to test whether there were potential non-sedimentary $CH_4$ sources. Anaerobic microsites in dense stands of submerged macrophytes have previously been shown to provide suitable substrate for methanogenic archaea and are a habitat for several aquatic microorganism that produce $CH_4$ via classic methanogenic pathways in an otherwise aerobic water column[58]. Indeed, 16 S rRNA gene amplicon sequencing of algal and/or organic matter debris in the macroalgae habitat confirmed the presence of methanogenic archaea classified as family Methanobacteriaceae, and genera *Methanocorpusculum*, *Methanomethylophilus*, and *Methanobrevibacter* (Supplementary Fig. 7). Together, these methanogens can use various substrates for methanogenesis, such as $CO_2$, acetate, and methylated compounds[59], suggesting a possible involvement of divergent $CH_4$ metabolisms within microsites of *Fucus vesiculosus* habitats that could act as sources of non-sedimentary $CH_4$ production. Importantly, methylotrophic methanogenesis can proceed in saline to hypersaline environments with high ambient sulfate concentrations[50,60,61] and is, thus, expected to also play an important role in coastal environments with higher salinity compared to the brackish waters of the Baltic Sea.

Lastly, we expanded our surface water $CH_4$ concentration measurements to a macroalgae-dominated area in the Gulf of Finland, 400 km east of our initial field site, to test whether: (1) measurements of high surface water $CH_4$ concentrations (Supplementary Table 4) in the macroalgae habitat of our main study area also occur at other locations; and (2) non-sedimentary $CH_4$ sources within coastal macroalgae habitats are sufficient to increase surface water $CH_4$ concentrations to a value above open water control observations. For this reason, measurements in the Finnish archipelago were performed in dense stands of *Fucus vesiculosus* that grew exclusively on hard bottom substrates surrounding a small rock island (Supplementary Fig. 8a). No sediment pockets underlying the macroalgae canopy were observed at this site. The results were compared to $CH_4$ concentrations measured in two open water control sites in 40–60 m distance to the island (Supplementary Fig. 8a). Results of a one-way ANOVA and subsequent Tukey's HSD Test for multiple comparisons (Supplementary Fig. 8b) found that the mean value of surface water $CH_4$ concentrations was significantly higher in the macroalgae site (mean = 56.69 nmol/L $CH_4$, SD = 0.76, *n* = 70) compared to the two open water locations (i.e., mean = 50.41 nmol/L $CH_4$, SD = 1.48, *n* = 105, *p* < 0.0001, 95% C.I. = [−6.95, −5.61]; and mean = 49.73 nmol/L $CH_4$, SD = 2.48, *n* = 119, *p* < 0.0001, 95% C.I. = [6.31, 7.62]). These results provide evidence for 1)

the occurrence of elevated surface water $CH_4$ concentrations across macroalgae habitats in various geographic locations and 2) local $CH_4$ production within macroalgae habitats even without underlying sediments.

While it becomes evident from the data that macroalgae habitats can produce and emit $CH_4$ locally – with possible sources from sediment pockets underlying the canopy, and/or anoxic organic matter microsites within dense stands that harbor methanogenic archaea – other indirect sources may also contribute to $CH_4$ formation in these oxic waters. For example, the production of $CH_4$ precursors and facilitation of their bacterial breakdown or chemical conversion have also been proposed[58] and warrant further investigations on the individual contribution of various sources and pathways of $CH_4$ production in macroalgae habitats.

## Accounting for $CH_4$ emissions from a wide range of coastal environments is crucial to inform efforts addressing climate change mitigation

Identifying the locations and mechanisms responsible for changing global atmospheric $CO_2$ and $CH_4$ is still a critical challenge for predicting future interactions between the carbon cycle and climate. The role of vegetated coastal ecosystems as a climate mitigation tool has attracted attention worldwide, with many countries pledged to use such systems as part of their nationally determined GHG inventories. However, traditional views of the radiative balance of coastal ecosystems often fail to capture some of the complexity of how these systems can impact the climate. Specifically, despite growing evidence for widespread $CH_4$ emissions from coastal ecosystems[5,6], the contribution to the net atmospheric GHG exchange remains unknown for many such environments and creates challenges for developing informed climate mitigation strategies[9].

Here, we showed that $CH_4$ emissions have to be measured in conjunction with sea-air $CO_2$ exchange to comprehensively evaluate the direction and magnitude of carbon-based GHG exchange with the atmosphere[21] and evaluate the net radiative balance of globally important coastal ecosystem. While autotrophic fucoid seaweed and mixed vegetation communities assessed in this study may efficiently remove $CO_2$ from the atmosphere during most of the year (Fig. 1b), concurrent $CH_4$ emissions (Fig. 1a) offset up to one-third of the net carbon sink capacity attributed to atmospheric $CO_2$ uptake over an annual cycle (Fig. 2a, b). Interest in financing coastal restoration or afforestation through the sale of carbon offset credits[62] adds urgency to including counterbalancing $CH_4$ emissions to evaluate the atmospheric carbon removal function correctly. Notably, macroalgae habitats are proclaimed to be Earth's largest vegetated coastal biome with exceptionally high rates of net primary production[34], and are, thus, already part of such carbon offset schemes[63] but have no reported in situ $CH_4$ emission rates identified to date.

Therefore, we recommend including direct sea-air $CH_4$ alongside $CO_2$ flux measurements from a wider range of vegetated and unvegetated coastal environments in future assessments as a necessary step to: (1) improve baselines for tracking emission trends, which currently rely primarily on carbon stock changes[23] or productivity measurements using oxygen; (2) identify factors and processes that increase GHG emissions from coastal environments, as, for example, $CH_4$ emissions are highly sensitive to temperature[39] and anthropogenic perturbations[6]; (3) integrate long-term continuous observations with process-based biogeochemical models to reduce uncertainties in estimating coastal carbon budgets as well as their climate effects; and (4) appreciate the high spatiotemporal heterogeneity in vegetation cover and functions related to carbon turnover of coastal ecosystems, which presently challenges GHG flux estimates[9,40].

In conclusion, our simultaneous high-resolution sea-air $CO_2$ and $CH_4$ flux measurements show that $CH_4$ emissions can offset one-third

of the carbon sink capacity attributed to atmospheric $CO_2$ uptake over an annual cycle across highly productive macroalgae and mixed vegetation coastal ecosystems in a northern temperate region. Net atmospheric $CO_2$ uptake still outweighs $CO_2$-eq. $CH_4$ emissions – that is, these habitats exert a net cooling impact over centurial timescales. This net radiative forcing benefit contrasts surrounding unvegetated sediment areas, which act as net atmospheric $CO_2$ and $CH_4$ source. Thus, the conservation and restoration of vegetated coastal ecosystems is advocated because it may effectively remove $CO_2$ from the atmosphere and reduce the adverse effects of climate change. However, the coastal atmospheric carbon sink capacity may be smaller than currently established, as spatially and temporally resolved $CH_4$ emissions remain unaccounted for in many coastal environments. Knowledge on habitat-specific $CH_4$ production pathways[50], including potential oxic $CH_4$ production mechanisms[58], and data on $CH_4$ emissions from various coastal ecosystems are, thus, needed to inform efforts addressing climate change with the net potential of coastal ecosystems to act as atmospheric carbon sinks.

## Methods

### Study area

The study was conducted on the island of Askö in the Baltic Sea (58°49′15.4″N 17°38′08.8″E) in 2020. Three distinct shallow (<4 m water depth) coastal habitats were identified according to their dominant type of substrate and vegetation: (1) Mixed-vegetated communities of vascular plants and algae on sediments (hereafter 'mixed vegetation' habitat; (2) macroalgae on rocks with pockets of sediments (hereafter 'macroalgae' habitat), and (3) surrounding soft sediments with marginal macrovegetation cover (hereafter 'bare sediments'). Each habitat was assessed visually, and the percent cover of the underlying substrate and macrovegetation was recorded within a 5 m radius. Benthic surveys were repeated in April and September 2020. The habitat with 'mixed vegetation' was characterized by coarse sediments with 60–90% total vegetation cover. The vegetation was dominated in equal parts by vascular plants (e.g., *Phragmites australis*, *Stuckenia pectinata*, and *Ruppia spiralis*) and benthic algae (e.g., *Chara aspera* and heterogenous assemblages of filamentous algae). The 'macroalgae' habitat was situated on rocks and boulders with pockets of permeable sediments with 80–95% total vegetation cover comprised of the macroalgae *Fucus vesiculosus*, and *Ulva* spp., the encrusting *Hildenbrandia rubra*, and various filamentous algae. No vascular plants were identified in this habitat. The surrounding 'bare sediment' habitat had 7–10% total vegetation, of which were mainly dislodged *F. vesiculosus* and filamentous algae. The habitats were fully submerged at all times due to the absence of tides in this region of the Baltic Sea[64]. The average of measured salinities in the studied habitats ranged from 6.6 to 7.0 over the course of the year, and, thus, reflected brackish water conditions typical for the central Baltic Sea with freshwater inflows from land and limited salt water inflows from the Danish straits. However, locally at the study site on the island in the outer Stockholm archipelago, there were no major freshwater inputs from rivers or streams, which is reflected by relatively constant salinity throughout the measurement period. Additional assessments were conducted in the Finnish archipelago in the Baltic Sea (59°50′30.8″N 23°15′01.0″E) in October 2021. Details about this site and the measurements performed are outlined in the section "$CH_4$ concentration measurements within and outside of a macroalgae-dominated habitat of the Finnish archipelago".

### Quantification of surface water and atmospheric $CO_2$ and $CH_4$

The partial pressures of surface water and atmospheric $CO_2$ and $CH_4$ in the three habitats were quantified during four measurement periods over an annual cycle (i.e., spring = 18–29 May; summer = 06–11 July; fall = 22 October to 2 November; and winter = 30 November to 08 December 2020). $CO_2$ and $CH_4$ concentrations were measured using a

fast-response automated gas equilibrator and cavity ring-down spectrometer (CRDS; model G2201-i, Picarro Inc.) according to protocols outlined in Roth et al. (2022)[40]. Briefly, surface water (at around 30 cm depth) was drawn in by a submersible pump from a floating ponton that was positioned over the habitats, and the water was transferred to a showerhead equilibrator (1 L headspace volume). From the equilibrator, a continuous air loop was linked to the CRDS, where $CO_2$ and $CH_4$ were measured in the dried gas stream for 35 minutes, followed by gas measurements of ambient air for 10 minutes (i.e., one complete cycle was 45 minutes). These measurement cycles ran continuously during the measurement periods mentioned above and the ponton with the submersible pump was moved between the defined habitats every 24 h. Concentrations measured at 1 Hz frequency were averaged and logged every 10 s. The recorded data were filtered by removing data from the transition period between stations and ambient air and water measurements due to the response time of CRDS to sharp changes in concentrations. Data was also removed during improper functioning (e.g., low water flow).

## Environmental data

Alongside CRDS measurements, several environmental and meteorological variables were recorded. Before the showerhead equilibrator, surface water was pumped into a flow-through chamber, where ancillary data (salinity, temperature) were measured with every CRDS measurement using a thermosalinograph (Seabird TSG 45). Surface water temperature, pH, and dissolved oxygen concentrations at the point of water intake were logged every 15 min using a multiparameter sonde (model EXO2, YSI) that was calibrated prior to each deployment. Wind data observations (wind speed and direction) and air temperature were obtained from a Metek uSonic-3 heated 3D sonic anemometer, and a Vaisala HMP155 shielded temperature probe mounted on a 1.5 m high meteorological mast. The mast was located at the waterline, ~400 m to the northwest of the sampled habitats. Mean winds were adjusted to a 10 m reference height assuming a logarithmic profile with neutral stability:[65]

$$U_{10} = U + \left(\frac{u^*}{\text{kappa}}\right) \times \log(\frac{10}{\text{zu}}) \qquad (1)$$

where $U$ is the measured wind speed at height zu, $u^*$ is the measured friction velocity by the 3D sonic anemometer, and kappa is the von Karman constant (0.4). Environmental data as well as surface water $CO_2$ and $CH_4$ concentrations are presented in Supplementary Table 4.

## Sea-air flux computation of $CO_2$ and $CH_4$

The sea-air flux ($F$) of $CO_2$ or $CH_4$ is calculated as:

$$F = k \times K_0 \times (\text{pGas}_\text{sea} - \text{pGas}_\text{air}) \qquad (2)$$

where $k$ (m s$^{-1}$) is the gas transfer velocity, $K_0$ (mol m$^{-3}$ atm$^{-1}$) is the aqueous-phase solubility of the respective gas, and pGas$_\text{sea}$ and pGas$_\text{air}$ are the measured partial pressures of $CO_2$ or $CH_4$ in the near-surface water and in the air, respectively. The solubilities were determined from Weiss 1974[66] for $CO_2$ and Wiesenburg and Guinasso 1979[67] for $CH_4$ as:

$$\ln \beta = A1 + A2\left(\frac{100}{T}\right) + A3\ln\left(\frac{T}{100}\right) + S\left[(B1 + B2(T/100) + B3(T/100)^2\right] \quad (3)$$

where $\beta$ is the dimensionless (mL of gas dissolved per mL of $H_2O$) Bunsen solubility coefficient, A1, A2, A3, and B1, B2, and B3 are constants, T is the measured water temperature (K) and $S$ the measured salinity. Assuming $CH_4$ behaves as an ideal gas, $K_0$ is related to $\beta$ in the above formula by $K_0 = \beta (R \times T_\text{STD})^{-1}$, where $R$ (m$^3$ atm K$^{-1}$ mol$^{-1}$) is the ideal gas constant and $T_\text{STD}$ (K) is the standard temperature in Kelvin.

The gas transfer velocity ($k$) used is that determined by Wanninkhof[68], as:

$$k = 0.251 \times U^2 \times \left(\frac{\text{Sc}_\text{balticsea}}{660}\right)^{-0.5} \qquad (4)$$

where $U$ is the wind speed (m s$^{-1}$) at 10 m height and Sc$_\text{balticsea}$ is the Schmidt number at the measurement site, which is dependent on temperature, salinity, and gas molecule. Sc was corrected for the corresponding temperature that was measured simultaneously with partial pressures the gases according to coefficients taken from Wanninkhof[68]. Further, the Schmidt number for Baltic Sea brackish water (i.e., Sc$_\text{balticsea}$) with measured salinity (S$_\text{balticsea}$) was calculated by interpolation of Sc for fresh water (salinity 0‰) and seawater (salinity 35‰) following refs. 69 and [70]:

$$\text{Sc}_\text{balticsea} = \frac{(\text{Sc}_\text{seawater} - \text{Sc}_\text{freshwater}) \times S_\text{balticsea}}{35} \times \text{Sc}_\text{freshwater} \quad (5)$$

Other variables (e.g., currents, waves, water depth) can also be used to predict $k$ in coastal environments, but the studied location does not have any significant permanent or tidal currents, and estuarine models may not provide better results for our setting. Further, Lundevall-Zara et al.[38] tested other wind-based $k$ models in similar habitats of the same location and concluded that calculated average $k$-values from different models were close to those of the Wanninkhof[68] relationship for the range of wind velocities encountered on the island of Askö. Fluxes were expressed in mg m$^{-2}$ day$^{-1}$ using the molecular weights of 44.01 g/mol and 16.04 g/mol for $CO_2$ and $CH_4$, respectively. First order estimates of annual fluxes (expressed in g $CO_2$ m$^{-2}$ y$^{-1}$ and g $CH_4$ m$^{-2}$ y$^{-1}$) are based on cumulative fluxes for each season.

We used the sustained-flux global warming potential (SGWP) as a greenhouse gas metric to describe the relative radiative impact of a standardized amount of gas over a defined time horizon[21]. Specifically, over a 100-year time horizon, the SGWP of $CH_4$ is 45 times greater than that of $CO_2$, on a mass basis, based on:

$$CO_2 - \text{eq}_{(CH_4)} = F_{(CH_4)} \times \text{SGWP}_{(CH_4)} \qquad (6)$$

where the $CO_2$-equivalent flux of methane ($CO_2$-eq$_{(CH4)}$) is the product of the flux ($F_{(CH4)}$) of $CH_4$ and its SGWP (i.e., 45) over the time horizon of 100 years.

## Sediment sampling

We collected multiple sediment cores for sediment biogeochemical and microbial assessments in May, July, August, and December 2020 (i.e., spring, summer, and winter; no sediment cores were taken in fall due to logistical constraints). In the mixed vegetation and macroalgae habitats, acrylic cores (50 cm length; 7 cm inner diameter) were pushed into the sediments by hand and the cores were subsequently plugged with rubber stoppers for transportation to the laboratory. While macroalgae grew on rocky substrates, we sampled sediment pockets underlying the canopy in-between rocks and boulders (exemplary picture in Supplementary Fig. 2a). Sediment was sampled using a Kajak core sampler with acrylic core liners (50 cm length; 7 cm inner diameter) for sampling in the bare sediment area. In total, we took three cores per habitat and sampling event, one each for $O_2$ microprofiles, sediment organic carbon contents, and DNA extraction.

Bottom water and sediment microprofiles for $O_2$ were performed with a 100-μm tip microsensor (OX-100, Unisense) that was maneuvered by a motor-driven micromanipulator (Unisense). Above the sediment, a water column layer of 4–6 cm was kept circulated by a gentle air flow to maintain a constant diffusive boundary layer during measurements. Signals were recorded and converted into concentrations with a four-channel multimeter (Unisense). Concentrations were

measured at a vertical resolution of 100 μm. Profiles were made in triplicates in each subsampled sediment core and were performed within few hours after sampling. The $O_2$ microsensor was calibrated using a two-point calibration procedure in $O_2$ saturated water conditions (100% $O_2$ air saturation) and inside the sediment (0% $O_2$ air saturation). We report oxygen penetration depth (OPD) as the depth where $O_2$ concentration became <1 μM[71].

One core from each habitat and sampling period was used for estimating sediment organic carbon contents. In the laboratory, the cores were sliced at 1 cm intervals in the first 10 cm, and in 2 cm intervals thereafter. Organic carbon content in 5 mL subsample from each slice was estimated as a gravimetric loss-on-ignition (LOI) after combustion at 550 °C for 12 h.

Sediment for DNA extraction were sampled from cores with pre-drilled holes at 1 cm intervals. One core was collected per habitat and time point, yielding a total of 193 samples. During sampling, the holes were covered with water resistant tape that was later lifted and a sterile 3 ml syringe (Henke-Ject) was inserted to sub-sample the sediment. The sediment was transferred to 15 ml centrifuge tubes (Sarstedt) and stored at −20 °C until DNA extraction.

### Sediment DNA extraction, 16S rRNA gene amplification, and sequencing

DNA was extracted from 0.25 g thawed and homogenized sediment from each sample using the DNeasy PowerSoil Pro Kit (QIAGEN) following the manufacturer's instructions. In addition, two blanks (only containing lysis buffer solution CD1) were extracted for DNA following the manufacturer's instructions. DNA quantity and quality were measured on a NanoDrop one spectrophotometer (Thermo Scientific). The DNA was stored at −80 °C until shipped to Novogene (Cambridge, UK) for PCR amplification, library preparation, and sequencing. The DNA concentration was normalized by Novogene according to their in-house company protocols. Amplification of the 16S rRNA gene V4 region was conducted with the primers 515 F[72] and 806 R[73], and library preparation was conducted using the NEBNext® Ultra™ II DNA Library Prep Kit with index adapters synthesized in-house by Novogene. The library was sequenced on the Illumina NovaSeq 6000 SP platform with a 2 × 250 bp paired-end which yielded 18.9 million read-pairs. See Supplementary Data 1 for a full list of sample names, fastq file names, sequences obtained before and after quality trimming, number of amplicon sequence variants (ASVs) constructed etc. The raw sequencing data has been uploaded to NCBI GenBank (https://www.ncbi.nlm.nih.gov/bioproject/) and can be accessed at BioProject PRJNA756121.

### Bioinformatics

The sequencing yielded on average of 97,077 read-pairs per sample (min: 30,288, max: 119,686). Illumina adapters were removed from the raw reads by using SeqPrep 1.2[74] with the parameters: -A AGATCGG AAGAGCACACGTCTGAACTCCAGTCA and -B AGATCGGAAGAGCGT CGTGTAGGGAAAGAGTGT. The sequences were then analyzed following the DADA2 pipeline[75] using the DADA2 1.21.0 package in R[76]. The following quality trimming parameters were used to remove primer sequences, low quality bases, and low quality reads: truncLen=c(240,240), maxEE=2, truncQ=2, maxN=0, rm.phix=TRUE, trimLeft=c(21, 22). The raw and filtered data were visualized as FastQC 0.11.9 reports using MultiQC 1.11[77,78] to ensure the filtering was successful. The error model was run using parameters: nread=1e6, MAX_CONSIST = 30; the merging step using minOverlap=10; and the chimera removal step using the parameters allowOneOff=TRUE and minFoldParentOverAbundance=4. The ASVs were annotated against the SILVA nr99 v138.1 database[79]. Singletons, chloroplasts, and mitochondria sequences were removed from the ASV table. Finally, ASVs only attributed to the blank samples and ASVs > 1000 counts in the blank samples were removed. The final dataset consisted of 18,717 ASVs and had on average 703 ASVs per sample (min: 162, max: 1155),

with an average of 51,837 read counts per sample (min: 16,453, max: 74,843). The data were analyzed as relative abundances (%) using the software Explicet 2.10.5[80]. The full list of the DADA2 results including ASVs with their partial 16S rRNA gene sequence, classified taxonomy, and read counts is available in Supplementary Data 1.

### Analysis of filamentous floating algal and/or organic matter debris associated with dense stands of macroalgae

Floating filamentous algal and/or organic matter debris associated with dense stands of macroalgae (exemplary pictures in Supplementary Fig. 2a, b) were collected in the macroalgae habitats of the main sampling site (58°49′15.4″N 17°38′08.8″E) in August 2020. Three samples were collected in 50 ml centrifuge tubes and flash frozen in liquid nitrogen and stored at −80 °C. DNA was extracted from 3 g homogenized material using the DNeasy PowerWater kit (Qiagen). The DNA was then handled and sequenced as mentioned in the section "Sediment DNA extraction, 16S rRNA gene amplification, and sequencing", except that the archaeal 16S rRNA gene V4–V5 region was amplified by the Novogene sequencing facility using primers Arch519F (CAGCCGCCGCGGTAA) and Arch915R (GTGCTC CCCCGCCAATTCCT)[81]. The delivered sequencing data had already been pre-trimmed for primers by Novogene and was analyzed with DADA2 using quality trim settings maxEE=2, truncQ=2, maxN=0, truncLen=c(215,215); error model nbases=1e8; merging minOverlap=10, maxMismatch=0; and chimera removal removeBimeraDenovo(method = "consensus"). The data were annotated against the SILVA NR99 v138.1 database. The final ASV counts were normalized as relative abundances (%). The full list of the DADA2 results including ASVs with their partial 16S rRNA gene sequence, classified taxonomy, and read counts is available in Supplementary Data 1.

### CH₄ concentration measurements within and outside of a macroalgae-dominated habitat of the Finnish archipelago

To test whether measurements of high surface water $CH_4$ concentrations (Supplementary Table 4) in the macroalgae habitat of our main study site could also be found in other shallow coastal locations dominated by macroalgae, we expanded $CH_4$ concentration measurements to a macroalgae-dominated area 400 km further east to our original field site. The measurements in the Finnish archipelago (59°50′30.8″N 23°15′01.0″E) were performed in an exposed, dense stand of *Fucus vesiculosus* that grew on hard bottom substrates without any apparent sediment pockets underlying the canopy. Specifically, we selected a rocky island and measured surface water $CH_4$ concentrations in the *Fucus vesiculosus* stands directly surrounding the island, and in "open water" control sites in 40 – 60 m distance to the island (Supplementary Fig. 8). Surface water $CH_4$ concentrations were measured according to the protocol outlined in the section "Quantification of surface water and atmospheric $CO_2$ and $CH_4$". Surface water $CH_4$ concentrations were recorded continuously for 1 h in each location (see Supplementary Fig. 8A: location "1" = within the dense macroalgae stands directly surrounding the island; "2" = open water control site; "3" = open water control site) between 09:00 – 12:00 on October 13, 2021. A one-way ANOVA with Tukey's HSD Test for multiple comparisons was performed to compare the effect of the three different locations on surface water $CH_4$ concentrations (results presented together with Supplementary Fig. 8).

### Data analysis

GHG flux data did not fulfill the requirements for general linear models (normal distribution of the dependent variables within groups, homogeneity of variances); thus, we performed the align-and-rank data for nonparametric factorial ANOVA procedure according to Wobbrock et al.[82]. After data preparation according to Wobbrock et al.[82], we performed two-way repeated measures ANOVAs in order to determine whether there was a significant interaction between the two

within subject factor variables "habitat" (i.e., the three habitat types 'macroalgae', 'mixed vegetation', and 'bare sediments') and "season" (i.e., sampling time-point in 'spring', 'summer', 'fall', and 'winter') on the dependent variables (i.e., '$CO_2$' and '$CH_4$' flux data), considering the individual subject identifier (i.e., habitat ID).

The 16S rRNA gene ASV data (normalized as relative abundance %) was used to construct non-metric multidimensional scaling (NMDS) plots based on the Bray-Curtis dissimilarity index and PERMANOVA (9999) tests using the software Past 4.07b[83].

## Data availability

All data needed to evaluate the conclusions in the paper are present in the paper and/or the Supplementary Materials. Source data for all $CH_4$, $CO_2$, and $CO_2$-eq. $CH_4$ fluxes (hourly and daily integrated), 16S rRNA gene ASV tables, and sediment organic carbon content and oxygen penetration profiles are provided with this paper. Raw sequencing data that support the findings of this study have been deposited in the NCBI GenBank (https://www.ncbi.nlm.nih.gov/bioproject/) with the accession code: BioProject PRJNA756121. Source data are provided with this paper.

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

## Acknowledgements
This research is part of the University of Helsinki and Stockholm University collaborative research initiative (CoastClim, www.coastclim.org; The Baltic Bridge initiative). We thank the staff at Stockholm Universities' field station Askö for their logistical support, and Joakim Hansen for the help with benthic macrophyte surveys. The study was funded by the Academy of Finland (Project ID 294853) and by the Walter and Andrée de Nottbeck Foundation to AN. Financial support was provided by the Swedish Research Council Formas to EB (grant no: 2020-02304). The bioinformatic analyses were enabled by resources in project SNIC 2022/22-405 provided by the Swedish National Infrastructure for Computing (SNIC) at UPPMAX, partially funded by the Swedish Research Council through grant agreement no. 2018-05973.

## Author contributions
F.R., C.H., and A.N. conceptualized the study; F.R., X.S., M.C.G., and C.H. developed GHG flux methodology; E.B. and F.N. developed microbial methodology; C.H., A.N., J.P., E.B., F.N., S.B., and V.B. provided resources and material; F.R., E.B., X.S., S.B., F.N., J.P., M.L.Z., and M.B. performed investigation; F.R., E.B., S.B., M.L.Z, C.H., and A.N. contributed to formal data analysis and data interpretation; F.R., E.B., C.H., and A.N. wrote the original draft; all authors reviewed and edited the manuscript.

## Funding

## Competing interests
The authors declare no competing interests.
