## [Peer Review File · Nature Communications]

Methane emissions offset atmospheric carbon dioxide uptake
in coastal macroalgae, mixed vegetation and sediment
ecosystemsReviewer #1 (Remarks to the Author):

This is a study that quantifies sea – air CO₂ and CH₄ fluxes in three coastal habitats composed of macroalgae, mixed vegetation and unvegetated sediments to determine their role as carbon sinks. The manuscript emphasizes the need of accounting for CH₄ alongside CO₂ fluxes, as part of the carbon that is captured by primary producers may be returned to the atmosphere as CH₄, which has a higher radiative efficiency than CO₂. The authors found that CH₄ emissions in these habitats offset their carbon sink capacity, being this offset variable between habitats and seasons. In the two autotrophic habitats studied, they found that CH₄ emissions can lower the greenhouse gas sink by one third over an annual cycle. Along with CO₂ and CH₄ fluxes, the study is supported with data to evaluate the microbial community, specifically the presence of methanogens in the sediments of the studied areas and the geochemical conditions that allow CH₄ production.

I found the research very interesting and with good quality, with a good methodological approach. My only concern is that only three very specific coastal habitats were studied: 1) rocks covered with macroalgae over a sediments substrate, 2) a mixed – vegetation community on sediments and 3) bare sediments with scarce vegetation. These three habitats were located in brackish waters areas, where salinity is very low (being the higher measured value of 7.0). Are these characteristics (the presence of sediments and low salinity) providing the conditions to a higher CH₄ production by methanogenic archaea? Can CH₄ production be so important in other coastal with higher salinities? I agree with the statement that CH₄ emissions should be accounted for in order to quantify the potential of coastal ecosystems as carbon sinks, and this work adds data regarding this issue, but the low variety of habitats studied prevent this work from completely reaching this conclusion. The scope of this study is accurately established in lines 280 – 283, and maybe it is not wide enough to fully support the main conclusion. Lines 267- 268 state that macroalgae habitats, recently classified as blue carbon systems, have no reported CH₄ emissions rates reported, and, after reading this manuscript, I fully agree with this statement. Therefore, I would have liked to see more macroalgae habitats included in this work.

Specific comments

Line 90: For the reasons previously stated, I think that these ecosystems cannot be described as globally prevalent.

Table 1: Figure 2 represents the data already included in Table 1, so I suggest including only one of the two in the main text. I would keep Figure 2 as it very effectively summarizes the main results and move Table 1 to Supplementary Material.

Figure 1: The scale of the y axis in Macroalgae in (B) has two decimals when in the other two habitats there is only one.

Figure 2 (A): The different scales of the y axis make the visual comparison between habitats difficult. I suggest using the same scale for the three habitats.

Lines 249 – 285: I would move this paragraph to the introduction.

Celso A. Hernández Díaz

Reviewer #2 (Remarks to the Author):

The paper by Roth et al. quantifies water-air CO₂ and CH₄ fluxes using high resolution continuous measurements in three coastal habitats (macroalgae, mixed vegetation, bare sediments) over 4 seasons. They present water-air CO₂ offsets by water-air CH₄ fluxes and find that offsets were greatest in fall in bare sediments (84%), followed by mixed vegetation (47%) and macroalgae (44%).

I very much enjoyed reading the paper by Roth et al., which is well written, contains great figures and tables, and statistical analysis. The methods have been used successfully in several previous studies (therefore not new) and as such are without any major flaws. The quality of data is technically sound, presented well and worthwhile publication. Especially the comparison of the three coastal habitats over different seasons. I do, however, find the study in context of the blue carbon discussion questionable. It remains controversial whether macroalgae should be included or rejected in the blue carbon framework (see Krause-Jansen et al. 2018). The authors state that the macroalgae site was mostly rocks with pockets of sediments. Most of the blue carbon (if any) potential would therefore be lateral export, which was not measured or considered in this study. The second habitat i.e. bare sediments cannot be put in the coastal vegetated blue carbon ecosystem category as they are obviously missing vegetation. The third habitat is mixed vegetation which makes it difficult to allocate effects to certain species. I would suggest re-thinking the title to better reflect the different coastal habitats. The most novel finding and center piece of this study/data, in my opinion, is the CH₄ emissions from the macroalgae habitat. Interesting is also that the studied coastal habitats are all fully submerged and non-tidal. Which could be a great comparison to previous published coastal blue carbon CO₂ and CH₄ studies that are mostly in tidal emergent coastal wetlands. Another comment would be that the water-air CO₂ flux cannot directly be translated to a carbon sink capacity. Any sea water has CO₂ uptake but you would have to provide evidence (and quantify how much) that this results in a long term (blue) carbon sink in local sediments or export. In this study, the authors measured the water-air flux and show that the CO₂ flux is (mostly) negative which means the surrounding water takes up CO₂. This means CO₂ removal from the atmosphere to the water column. But this is not blue carbon. What is the fate of CO₂ once it is in the water column? Some of it will be taken up by primary production (could be pelagic or benthic algae, vascular plants, or others), some may be buried in sediments, some may be exported. This needs to be discussed and ideally quantified, including carbonate chemistry. In fact, the authors found the greatest water-air CO₂ flux offsets by CH₄ in bare sediments in summer and fall. But it is unclear to me why this is then put in context with blue carbon burial offsets in vegetated coastal sediments in the discussion? Again, the data and study are great. It's the blue carbon story that doesn't sit quite right. I would suggest to simply compare CO₂ and CH₄ fluxes and remove the blue carbon/carbon sink offset discussion. I could also envisage that the focus of this manuscript shifts to a water-air CO₂ /CH₄ flux comparison of submergent vs emergent coastal wetlands. Finally, I find it an overstatement of the authors to state that their measurements are the first to resolve CH₄ over hourly, daily, or seasonal time scales in coastal environments (L117-118). There are several teams that have published continuous CO₂ and CH₄ in coastal (blue carbon) ecosystems. For example, see Santos et al. 2012, Maher et al. 2013, Call et al. 2014, and Call et al. 2019. And eddy covariance towers: Liu et al. 2020, Jha et al. 2014, Holm et al. 2016, Van Dam et al. 2021. Water-air CO₂ and CH₄ fluxes in coastal wetlands were reviewed in Rosentreter 2022.

Krause-Jensen, D. et al. Sequestration of macroalgal carbon: The elephant in the Blue Carbon room. *Biol. Lett.* 14, (2018).

Santos, I. R., Maher, D. T. & Eyre, B. D. Coupling automated radon and carbon dioxide measurements in coastal waters. *Environ. Sci. Technol.* 46, 7685–7691 (2012).

Maher, D. T. et al. Novel Use of Cavity Ring-down Spectroscopy to Investigate Aquatic Carbon Cycling from Microbial to Ecosystem Scales. *Environ. Sci. Technol.* 47, 12938–12945 (2013).

Call, M. et al. Spatial and temporal variability of carbon dioxide and methane fluxes over semi-diurnal and spring-neap-spring timescales in a mangrove creek. *Geochim. Cosmochim. Acta* 150, 211–225 (2015).

Call, M. et al. High pore-water derived CO₂ and CH₄ emissions from a macro-tidal

mangrove creek in the Amazon region. *Geochim. Cosmochim. Acta* 247, 106–120 (2019).

Liu, J. et al. Methane emissions reduce the radiative cooling effect of a subtropical estuarine mangrove wetland by half. *Glob. Chang. Biol.* 1–19 (2020)
doi:10.1111/gcb.15247.

Jha, C. S., Rodda, S. R., Thumaty, K. C., Raha, A. K. & Dadhwal, V. K. Eddy covariance based methane flux in Sundarbans mangroves, India. *J. Earth Syst. Sci.* 123, 1089–1096 (2014).

Holm, G. O. et al. Ecosystem Level Methane Fluxes from Tidal Freshwater and Brackish Marshes of the Mississippi River Delta: Implications for Coastal Wetland Carbon Projects. *Wetlands* 36, 401–413 (2016).

Van Dam, B. R. et al. Calcification-driven CO₂ emissions exceed “Blue Carbon” sequestration in a carbonate seagrass meadow. *Sci. Adv.* 7, 1–12 (2021).

Rosentreter, J. A. Chapter 6 - Water-air gas exchange of CO₂ and CH₄ in coastal wetlands. in *Estuarine and Coastal Sciences Series* (eds. Ouyang, X., Lee, S. Y., Lai, D. Y. F. & Marchand, C. B. T.) vol. 2 167–196 (Elsevier, 2022).

Reviewer #3 (Remarks to the Author):

This manuscript provides very interesting results in its field, in which the large role that CH₄ emissions can play in the context of climate change. Also, It encourages future studies to focus not only on CO₂ emissions.

To reach these conclusions, the authors have used a fairly comprehensive and robust methodology, with sufficient replications for a robust analysis. They have compiled a large amount of chemical and biological data, and have been able to interpret and present it with figures adequately, which are of great support for their assertions.

Therefore I consider it a good manuscript to publish in this journal, although I suggest some improvements:

1)Small corrections:

- Line 90 and 156: put "in situ" in italics.
- Line 419: change "µM" to "µm".

2)Comments on methodology:

- Section "DNA extraction, 16S rRNA gene amplification, and sequencing": I feel that more detail on amplification and sequencing is needed. What concentration of DNA did you use to amplify? did you use labels? which ones? which adapters did you illuminate?
- Data analysis" section: I would also need more detail for reproducibility of the analyses carried out with the 16S data. What data exactly did you use, maybe reads or absence or presence of species?

REVIEWER COMMENTS

Reviewer #1 (Remarks to the Author):

This is a study that quantifies sea – air CO₂ and CH₄ fluxes in three coastal habitats composed of macroalgae, mixed vegetation and unvegetated sediments to determine their role as carbon sinks. The manuscript emphasizes the need of accounting for CH₄ alongside CO₂ fluxes, as part of the carbon that is captured by primary producers may be returned to the atmosphere as CH₄, which has a higher radiative efficiency than CO₂. The authors found that CH₄ emissions in these habitats offset their carbon sink capacity, being this offset variable between habitats and seasons. In the two autotrophic habitats studied, they found that CH₄ emissions can lower the greenhouse gas sink by one third over an annual cycle. Along with CO₂ and CH₄ fluxes, the study is supported with data to evaluate the microbial community, specifically the presence of methanogens in the sediments of the studied areas and the geochemical conditions that allow CH₄ production.

I found the research very interesting and with good quality, with a good methodological approach.

My only concern is that only three very specific coastal habitats were studied: 1) rocks covered with macroalgae over a sediments substrate, 2) a mixed – vegetation community on sediments and 3) bare sediments with scarce vegetation. These three habitats were located in brackish waters areas, where salinity is very low (being the higher measured value of 7.0). Are these characteristics (the presence of sediments and low salinity) providing the conditions to a higher CH₄ production by methanogenic archaea? Can CH₄ production be so important in other coastal with higher salinities?

I agree with the statement that CH₄ emissions should be accounted for in order to quantify the potential of coastal ecosystems as carbon sinks, and this work adds data regarding this issue, but the low variety of habitats studied prevent this work from completely reaching this conclusion. The scope of this study is accurately established in lines 280 – 283, and maybe it is not wide enough to fully support the main conclusion. Lines 267- 268 state that macroalgae habitats, recently classified as blue carbon systems, have no reported CH₄ emissions rates reported, and, after reading this manuscript, I fully agree with this statement. Therefore, I would have liked to see more macroalgae habitats included in this work.

Response: Thank you very much for the positive feedback and constructive comments. In the following, we respond to all of the above concerns, as they all go in the same direction: Are the observations from the macroalgae habitat reproducible in other locations and under different conditions?

In the revised version of the manuscript, we thus present additional data showing that high surface water CH₄ concentrations result from divergent involvement of CH₄ metabolism specific to macroalgae habitats.

First: In addition to the already identified sediment pockets as sources for classical sedimentary CH₄ production, we added 16S rRNA gene amplicon sequencing of floating algae and/or organic matter debris associated with dense stands of macroalgae. The data identify this material as an additional source for local CH₄ production within the oxic water column. Importantly, the sequencing data provides evidence for the presence of methylotrophic methanogenesis. This indicates that CH₄ production can proceed under high ambient sulfate concentrations and is, thus, expected to play an important role in coastal environments with high salinity, even outside the brackish waters of the Baltic Sea (more details on the question regarding high salinity environments below). The new data is presented in the paragraph “Distinct microbial communities shape habitat-specific CH₄ dynamics” (lines 679 - 692) and the new Figure S7. In addition, we added a new figure to the supplementary material (Fig. S2) exemplifying locations from which samples for 16S rRNA gene amplicon sequencing were taken: i.e., sediment pockets underlying the canopy in between rocks and boulders, and floating algal and/or organic matter debris associated with dense stands of *Fucus vesiculosus*.

Second: We expanded measurements to another macroalgae-dominated area 400 km East of our initial field site. The data shows that high surface water CH₄ concentrations in macroalgae habitats are reproducible in other locations. Importantly, the habitat characteristics differed from our initial field site, as macroalgae grew exclusively on rocks and had no underlying sediments. Thus, the data confirms that non-sedimentary CH₄ sources are sufficient to increase surface water CH₄ concentrations in macroalgae habitats above open water control values. The data is presented in the paragraph “Distinct microbial communities shape habitat-specific CH₄ dynamics” (lines 694 – 718) and the new Figure S8.

CH₄ production in submerged vegetation has recently been discussed in the review “Potential role of submerged macrophytes for oxic methane production in aquatic ecosystems” (Hilt et al. 2022) – an important study that we now refer to in the discussion of our manuscript. As we understand that our work

cannot explain all processes involved, we added to the discussion that CH₄ production pathways need to be further explored in currently underrepresented vegetated coastal habitats.

Regarding the questions “Can CH₄ production be so important in other coastal with higher salinities?”: This is a valid point, which we missed to discuss in the initial version of the manuscript. Indeed, high salinity and the associated increase in sulfate availability can suppress methanogenesis, as sulfate-reducing bacteria outcompete methanogens for hydrogen. However, elevated CH₄ emissions are also observed in saline to hypersaline environments across a wide range of sulfate concentrations (e.g., Beversdorf et al. 2008; Mcgenity and Sorokin 2018; Schorn et al. 2022). Methane can be produced aerobically through methylphosphonate degradation, such that competition with sulfate-reducers is bypassed. Methylophilic methanogenesis can also proceed uninhibited by high sulfate concentrations (Mcgenity and Sorokin 2018; Oremland and Polcin 1982). Our new data provides evidence for the involvement of methylophilic methanogenesis within macroalgae habitats (please refer to lines 690 – 692).

Literature:

Mcgenity T, Sorokin D. Methanogens and methanogenesis in hypersaline environments. Biogenesis of hydrocarbons. Springer International Publishing, New York, NY, USA; 2018. p. 1–27.

Oremland RS, Polcin S. Methanogenesis and sulfate reduction: competitive and noncompetitive substrates in estuarine sediments. *Appl Environ Microbiol.* 1982;44:1270–6.

Repeta DJ, Ferrón S, Sosa OA, Johnson CG, Repeta LD, Acker M, et al. Marine methane paradox explained by bacterial degradation of dissolved organic matter. *Nat Geosci.* 2016;9:884–7.

Karl DM, Beversdorf L, Björkman KM, Church MJ, Martinez A, Delong EF. Aerobic production of methane in the sea. *Nat Geosci.* 2008;1:473–8.

Changes to the manuscript:

The section “Distinct microbial communities shape habitat-specific CH₄ dynamics” now follows a three-point line of evidence starting with the sediment microbial work (already presented in the first version of the manuscript), followed by the floating filamentous algae and/or organic matter debris, and the expansion to another location. Please refer to lines 483 – 707:

“High rates of CH₄ emissions have been ascribed to mixed-vegetated (28, 44) and surrounding depositional (45, 46) areas with organic matter-rich sediments. The high rates of CH₄ emissions from the macroalgae habitats in our study are therefore intriguing because of the predominance on rocky hard-bottom substrates. To examine whether the high surface water CH₄ concentrations (Table S4) in this habitat-type were a consequence of lateral transport of dissolved CH₄ from neighboring habitats or a unique feature of this particular location, we performed three additional assessments:

First, we sampled bottom substrates of all habitats across seasons to evaluate the local geochemical and microbial potential for CH₄ production. We found organic matter-rich anoxic sediments, with similar organic carbon contents (3 – 5%) below 10 cm depth in all habitats and months (Fig. S1; selected results from summer are shown in Fig. 3A). While macroalgae grew on rocks, organic matter-rich sediment was present in sediment pockets underlying the canopy, between rocks and boulders (exemplary picture in Fig. S2A). Local organic carbon-rich deposits with anoxic conditions may be conducive to methanogenesis (13). The presence of methanogens in these sediments was confirmed with 16S rRNA gene amplicon sequencing. However, the three habitats showed distinctively different microbial community structures in the top 5 cm of the sediment during all four seasons (Bray-Curtis dissimilarity analyses, PERMANOVA, P = 0.0001; Fig. S3 and S4; example results from summer are shown in Fig. 3B), and methanogenic archaea were detected at all sites (Fig S5; exemplifying results from summer shown in Fig. 3C). Some of the retrieved 16S rRNA gene sequences in the mixed-vegetated and macroalgae systems were affiliated with classical methanogenic orders within the phylum Methanomicrobiales (i.e., using hydrogen to produce CH₄) and methylophilic Methanosarcinales (i.e., using one-carbon compounds such as methanol) (Fig. S5; selected results from summer are shown in Fig. 3C) (47). We also found unclassified sequences within the archaeal phylum Crenarchaeota (including the class Bathyarchaeia) in the macroalgae habitat (Fig. S5). Some of these archaea encode the universal phylogenetic marker gene mcrA used to identify methanogenic microorganisms (48). The bare sediments showed a prevalence of methylophilic methanogens of the phylum Methanomassiliicoccales near the sediment surface (Fig. S5; results from summer shown in Fig. 3b), which are major CH₄ producers in mangroves systems (49). Finally, deeper oxygen penetration depths

(Fig. S1) in the mixed-vegetated habitat are consistent with a higher relative abundance of methane-oxidizing *Methylococcales* bacteria, suggesting a more efficient sedimentary CH_4 oxidation filter in this habitat compared to the macroalgae and bare sediment area (Fig. S6). Together, the data show that distinct methanogenic communities in vegetated and unvegetated habitats are likely involved in producing CH_4 in sediments or sediment pockets between rocks locally (30).

Second, we took samples of organic matter aggregates (floating filamentous algal and/or organic matter debris) associated with dense stands of macroalgae (Fig. S2A and S2B) to test whether there were potential non-sedimentary CH_4 sources. Anaerobic microsites in dense stands of submerged macrophytes have previously been shown to provide suitable substrate for methanogenic archaea and are a habitat for several aquatic microorganism that produce CH_4 via classic methanogenic pathways in an otherwise aerobic water column (50). Indeed, 16S rRNA gene amplicon sequencing of algal and/or organic matter debris in the macroalgae habitat confirmed the presence of methanogenic archaea classified as *Methanobacteriaceae*, *Methanocorpusculum*, *Methanomethylophilus*, and *Methanobrevibacter* (Fig. S7). Together, these methanogens can use various substrates for methanogenesis, such as CO_2 , acetate, and methylated compounds (51), suggesting a possible involvement of divergent CH_4 metabolisms within microsites of *Fucus vesiculosus* habitats that could act as sources of non-sedimentary CH_4 production. Importantly, methylotrophic methanogenesis can readily proceed under high ambient sulfate concentrations and is, thus, expected to play an important role in coastal environments with high salinity, even outside the brackish waters of the Baltic Sea (30, 52, 53).

Lastly, we expanded our surface water CH_4 concentration measurements to a macroalgae-dominated area in the Gulf of Finland, 400 km east of our initial field site, to test whether: 1) measurements of high surface water CH_4 concentrations (Table S4) in the macroalgae habitat of our main study area also occur at other locations; and 2) non-sedimentary CH_4 sources within coastal macroalgae habitats are sufficient to increase surface water CH_4 concentrations to a value above open water control observations. For this reason, measurements in the Finnish archipelago were performed in dense stands of *Fucus vesiculosus* that grew exclusively on hard bottom substrates surrounding a small rock island (Fig. S8A). No sediment pockets underlying the macroalgae canopy were observed at this site. The results were compared to CH_4 concentrations measured in two open water control sites in 40 – 60 m distance to the island (Fig. S8A). Results of a one-way ANOVA and subsequent Tukey's HSD Test for multiple comparisons (Fig. S8B) found that the mean value of surface water CH_4 concentrations was significantly higher in the macroalgae site (mean = 56.69 nmol/L CH_4 , SD = 0.76, n = 70) compared to the two open water locations (i.e., mean = 50.41 nmol/L CH_4 , SD = 1.48, n = 105, $p < 0.0001$, 95% C.I. = [-6.95, -5.61]; and mean = 49.73 nmol/L CH_4 , SD = 2.48, n = 119, $p < 0.0001$, 95% C.I. = [6.31, 7.62]). These results provide evidence for 1) the occurrence of elevated surface water CH_4 concentrations across macroalgae habitats in various geographic locations and 2) local CH_4 production within macroalgae habitats even without underlying sediments.

While it becomes evident from the data that macroalgae habitats can produce and emit CH_4 locally – with possible sources from sediment pockets underlying the canopy, and/or anoxic organic matter microsites within dense stands that harbor methanogenic archaea – other indirect sources may also contribute to CH_4 formation in these oxic waters. For example, the production of CH_4 precursors and facilitation of their bacterial breakdown or chemical conversion have also been proposed (50) and warrant further investigations on the individual contribution of various sources and pathways of CH_4 production in macroalgae habitats.”

In addition, we added the below figures to the supplementary material:

Figure S2. Exemplary picture from the macroalgae habitat dominated by *Fucus vesiculosus* (A) and ex situ close-up of *Fucus vesiculosus* with associated floating algal and/or organic matter debris. Depicted in the pictures are exemplary locations from which additional samples for 16S rRNA gene amplicon sequencing were taken: i.e., sediment pockets underlying the canopy in-between rocks and boulders, and floating filamentous algal and/or organic matter debris associated with dense stands of *Fucus vesiculosus*. Picture in A) taken by Alf Norkko; picture in B) taken by Florian Roth.

A) Methanogenic archaea (% of archaeal community)

Figure S7. Sequencing results from organic matter aggregates (i.e., floating filamentous algal and/or organic matter debris) associated with dense stands of *Fucus vesiculosus*. Floating filamentous algal and/or organic matter debris (exemplary pictures in Fig. S2A and S2B) were collected from within dense stands of *Fucus vesiculosus* (i.e., the macroalgae habitat) in August 2020. A) shows the relative abundance (% of all archaea) of methanogenic archaea. The figure shows the results from sequencing of the archaeal 16S rRNA gene V4–V5 region and the lowest level of classified taxonomy. The x-axis shows the relative abundance (%) of all archaea, while the color legend shows methanogenic archaea detected in the samples with “Other archaea” denoting non-methanogenic archaea. B) shows the detailed relative abundance % of methanogenic archaea for the same three samples plotted as a heatmap.

Figure S8. (A) Aerial image of the sampling sites around a rocky island in the Finnish archipelago (59°50'30.8"N 23°15'01.0"E) surrounded by dense stands of *Fucus vesiculosus* (brown patches) and (B) results of surface water CH₄ concentration measurements at the different locations indicated in (A). Location “1” refers to within the dense macroalgae stands directly surrounding the island, while “2” and “3” are open water control sites. The concentration was recorded for approximately 1h continuously at each location between 09:00 – 12:00 on October 13, 2021. A one-way ANOVA was performed to compare the effect of the three different locations on surface water CH₄ concentrations. There was a statistically significant difference in mean surface water CH₄ concentrations between at least two groups (F(2, 291) = [348.2133], $p < 0.0001$). Tukey’s HSD Test for multiple comparisons found that the mean value of surface water CH₄ concentrations was significantly different between location 1 (mean = 56.69, SD = 0.76, n = 70) and 2 (mean = 50.41, SD = 1.48, n = 105) ($p < 0.0001$, 95% C.I. = [-6.95, -5.61]) and location 1 and 3 (mean = 49.73, SD = 2.48, n = 119) ($p < 0.0001$, 95% C.I. = [6.31, 7.62]). There was no statistically significant difference in mean surface water CH₄ concentrations between location 2 and 3 ($p = 0.167$). Abbreviations: ns = not significant. Picture in (A) taken by Alf Norkko.

Specific comments

Line 90: For the reasons previously stated, I think that these ecosystems cannot be described as globally prevalent.

Response: Please refer to our extensive response above. Especially macroalgal habitats are believed to be the most extensive and productive of all coastal vegetated ecosystems (see <https://doi.org/10.1111/geb.13515> and <https://doi.org/10.1038/s41597-022-01554-5>). We refer to these recent reviews at various instances in the revised version of the manuscript.

Table 1: Figure 2 represents the data already included in Table 1, so I suggest including only one of the two in the main text. I would keep Figure 2 as it very effectively summarizes the main results and move Table 1 to Supplementary Material.

Response: We have moved Table 1 to the Supplementary Material (now Table S1). Also, to maximise the reproducibility of data, we provide a file containing the raw data underlying reported means/errors in bar charts and tables in a folder ‘Source Data’, which will be referred to in the “Data Availability” section upon publication.

Figure 1: The scale of the y axis in Macroalgae in (B) has two decimals when in the other two habitats there is only one.

Response: We have adjusted the y-axis scales according to the comment below. All habitats are now on the same y-axis scale and have the same decimals. New figure:

Figure 1. Hourly sea-air CO₂ (A) and CH₄ (B) fluxes across four seasons in three coastal ecosystems. Values are means \pm standard error. Positive fluxes refer to an efflux from the water to the atmosphere (source), while negative fluxes depict an uptake of atmospheric GHGs (sink). In situ continuous (1Hz) measurements were averaged to 15 min intervals and binned in 2-hour blocks for graphical representation. Daily integrated net sea-air fluxes of CO₂ and CH₄ across seasons and habitats are presented in the text and Table S1.

Figure 2 (A): The different scales of the y axis make the visual comparison between habitats difficult. I suggest using the same scale for the three habitats.

Response: Thank you for the helpful suggestion. We have changed the scales of the y-axis in Figure 1 (A) and (B) to ease the visual comparison between habitats. The new figure is embedded in the revised version of the manuscript. We have also tried to adjust the y-axis scales of Figure 2 (A). However, the data range in this figure is too large (i.e., from -800 to 400) and varied across habitats (minimum value of -100 and -600 in the bare sediments and mixed-vegetated habitat, respectively) that a uniform y-axis scale across the habitats would prevent readability of the data.

Lines 249 – 258: I would move this paragraph to the introduction.

Response: Parts of this section have been removed entirely while other bits are reformulated due to the requested changes by reviewer 2 (i.e., less focus on Blue Carbon, stronger focus on rates and mechanisms of GHG exchange). See lines 833 – 841.

Reviewer #2 (Remarks to the Author):

The paper by Roth et al. quantifies water-air CO₂ and CH₄ fluxes using high resolution continuous measurements in three coastal habitats (macroalgae, mixed vegetation, bare sediments) over 4 seasons. They present water-air CO₂ offsets by water-air CH₄ fluxes and find that offsets were greatest in fall in bare sediments (84%), followed by mixed vegetation (47%) and macroalgae (44%).

I very much enjoyed reading the paper by Roth et al., which is well written, contains great figures and tables, and statistical analysis. The methods have been used successfully in several previous studies (therefore not new) and as such are without any major flaws. The quality of data is technically sound, presented well and worthwhile publication. Especially the comparison of the three coastal habitats over different seasons.

Response: We thank the reviewer for the positive feedback.

I do, however, find the study in context of the blue carbon discussion questionable. It remains controversial whether macroalgae should be included or rejected in the blue carbon framework (see Krause-Jansen et al. 2018). The authors state that the macroalgae site was mostly rocks with pockets of sediments. Most of the blue carbon (if any) potential would therefore be lateral export, which was not measured or considered in this study. The second habitat i.e. bare sediments cannot be put in the coastal vegetated blue carbon ecosystem category as they are obviously missing vegetation. The third habitat is mixed vegetation which makes it difficult to allocate effects to certain species. I would suggest re-thinking the title to better reflect the different coastal habitats.

Response: We agree with the reviewer and have toned down all discussion around “Blue Carbon”. Instead, in the revised version, the primary focus is on local water-to-atmosphere gas exchange. In addition, we have expanded the work to identify mechanisms by which high CH₄ emissions are sustained in various habitats – particularly in the macroalgae areas. Identifying the locations and mechanisms responsible for changing atmospheric CO₂ and CH₄ is important to predicting future interactions between the carbon cycle and climate. Please refer to the two comments below (i.e., CH₄ emissions from macroalgae; discussion around the carbon sink capacity) for more details on what has been changed in and/or added to the manuscript.

The most novel finding and centre piece of this study/data, in my opinion, is the CH₄ emissions from the macroalgae habitat. Interesting is also that the studied coastal habitats are all fully submerged and non-tidal. Which could be a great comparison to previous published coastal blue carbon CO₂ and CH₄ studies that are mostly in tidal emergent coastal wetlands.

Response: Thank you very much for this encouraging comment, which is in line with reviewer 1. Indeed, the most striking result of our study is the high CH₄ emissions from the macroalgae habitat. In the revised version, we have thus expanded on this matter and added additional data showing that high surface water CH₄ concentrations result from divergent involvement of CH₄ metabolism specific to macroalgae habitats and that high surface water CH₄ concentrations in macroalgae habitats are reproducible in other locations.

Specifically, we added:

- 16S rRNA gene amplicon sequencing of floating algae and/or organic matter debris associated with dense stands of macroalgae. The data identify this material as an additional source for local CH₄ production within the oxic water column.
- Data from another macroalgae-dominated area 400 km East of our initial field site. The results show that high surface water CH₄ concentrations in macroalgae habitats are reproducible in other locations, even without sediment pocks underlying the canopy.

We kindly ask you to refer to our response to the first comment of reviewer one, in which we detail all the changes and additions (including three new figures) to the manuscript.

Another comment would be that the water-air CO₂ flux cannot directly be translated to a carbon sink capacity. Any sea water has CO₂ uptake but you would have to provide evidence (and quantify how much) that this results in a long term (blue) carbon sink in local sediments or export. In this study, the authors measured the water-air flux and show that the CO₂ flux is (mostly) negative which means the surrounding water takes up CO₂. This means CO₂ removal

from the atmosphere to the water column. But this is not blue carbon. What is the fate of CO₂ once it is in the water column? Some of it will be taken up by primary production (could be pelagic or benthic algae, vascular plants, or others), some may be buried in sediments, some may be exported. This needs to be discussed and ideally quantified, including carbonate chemistry. In fact, the authors found the greatest water-air CO₂ flux offsets by CH₄ in bare sediments in summer and fall. But it is unclear to me why this is then put in context with blue carbon burial offsets in vegetated coastal sediments in the discussion? Again, the data and study are great. It's the blue carbon story that doesn't sit quite right. I would suggest to simply compare CO₂ and CH₄ fluxes and remove the blue carbon/carbon sink offset discussion. I could also envisage that the focus of this manuscript shifts to a water-air CO₂ /CH₄ flux comparison of submergent vs emergent coastal wetlands.

Response: We agree with the reviewer on this point. In the revised version of the manuscript, we refrain from comparing our results to any "Blue Carbon" sink capacity but rather focus on the CO₂ and CH₄ fluxes, the mechanisms sustaining them, and the resulting GHG budget. When talking about counterbalancing methane emissions, we now refer to an offset in the atmospheric carbon dioxide uptake in macroalgae and mixed-vegetated coastal ecosystems.

This change in notion is reflected throughout the manuscript:

- The new title of the study is: "Unrecognized methane emissions offset atmospheric carbon dioxide uptake in macroalgae and mixed-vegetated coastal ecosystems"
- We removed all reference to "Blue Carbon burial" from the introduction but instead focus on air-sea gas exchange
- In all instances, we now talk about an "offset in the carbon sink attributed to atmospheric CO₂ uptake" rather than just an offset in the "carbon sink capacity"
- We removed the reference to "Blue Carbon" from the discussion and focused on identifying the locations and mechanisms responsible for changing atmospheric CO₂ and CH₄ as a critical challenge to predicting future interactions between the carbon cycle and climate (lines 833 – 841).

Finally, I find it an overstatement of the authors to state that their measurements are the first to resolve CH₄ over hourly, daily, or seasonal time scales in coastal environments (L117-118). There are several teams that have published continuous CO₂ and CH₄ in coastal (blue carbon) ecosystems. For example, see Santos et al. 2012, Maher et al. 2013, Call et al. 2014, and Call et al. 2019. And eddy covariance towers: Liu et al. 2020, Jha et al. 2014, Holm et al. 2016, Van Dam et al. 2021. Water-air CO₂ and CH₄ fluxes in coastal wetlands were reviewed in Rosentreter 2022.

Response: Thank you for raising this concern and highlighting the excellent work that has been done previously. It was not our intention to state that we were the first to resolve CH₄ over various time scales, but rather that we are not aware of studies making a direct comparison across different habitat-types. In order to avoid any misunderstanding, we have removed the paragraph from the revised version of the manuscript.

Reviewer #3 (Remarks to the Author):

This manuscript provides very interesting results in its field, in which the large role that CH₄ emissions can play in the context of climate change. Also, it encourages future studies to focus not only on CO₂ emissions. To reach these conclusions, the authors have used a fairly comprehensive and robust methodology, with sufficient replications for a robust analysis. They have compiled a large amount of chemical and biological data, and have been able to interpret and present it with figures adequately, which are of great support for their assertions. Therefore, I consider it a good manuscript to publish in this journal, although I suggest some improvements:

Response: Thank you very much for the positive feedback. We have addressed all suggestions for improvement in the below.

1) Small corrections:

- Line 90 and 156: put "in situ" in italics.
- Line 419: change "μM" to "µm".

Response: As we do not find any information in Nature's "Formatting Guide", we will leave the decision to change the formatting to the editing office upon acceptance of the manuscript. Regarding "μM" to "µm": μM is correct, corresponding to the molarity as the number of (micro) moles per liter, and is related to the O₂ concentration and not the O₂ penetration depth.

2) Comments on methodology:

- Section "DNA extraction, 16S rRNA gene amplification, and sequencing": I feel that more detail on amplification and sequencing is needed. What concentration of DNA did you use to amplify?

Response: This work was carried out by the sequencing facility as we send the extracted DNA to them. We have been in contact with Novogene, and they replied that this information was confidential. We have now clarified in the manuscript that the DNA concentration was normalized by Novogene according to their in-house company protocols.

Did you use labels? which ones?

Response: The library preparation was carried out by Novogene. We have also been in contact with them regarding this question. Novogene uses barcodes (indexes) during amplification and adapters (containing Rd 1 SP/Rd 2 SP, barcodes, and P5/P7 sequences) to prepare the libraries. The primers are synthesized by Novogene themselves. The kit they used during library preparation was NEBNext® Ultra™ II DNA Library Prep Kit. The libraries were then sequenced on the NovaSeq 6000 SP platform. In the revised version of the manuscript, we included the below section:

"The DNA was stored at -80°C until shipped to Novogene (Cambridge, UK) for PCR amplification, library preparation, and sequencing. The DNA concentration was normalized by Novogene according to their in-house company protocols. Amplification of the 16S rRNA gene V4 region was conducted with the primers 515F (63) and 806R (64), and library preparation was conducted using the NEBNext® Ultra™ II DNA Library Prep Kit with index adapters synthesized in-house by Novogene." (lines 1066 – 1070)

Which adapters did you illuminate?

Response: See the comments above for more details including what kit was used. The adapters included Rd 1 SP/Rd 2 SP, barcode indexes, and P5/P7 sequences. To further clarify what indexes were used, we have included the barcode index sequences in Supplementary Data 1.

- "Data analysis" section: I would also need more detail for reproducibility of the analyses carried out with the 16S data. What data exactly did you use, maybe reads or absence or presence of species?

Response: We have now clarified this in the data analysis section. It now reads: "The 16S rRNA gene ASV data (normalized as relative abundance %) was used to construct non-metric multidimensional scaling (NMDS) plots based on the Bray-Curtis dissimilarity index and PERMANOVA (9999) tests using the software Past 4.07b (73)." (lines 1149 – 1151)

Reviewer #1 (Remarks to the Author):

After reviewing the revised manuscript, I find that it has improved significantly, with new data and references regarding the major concerns that I had with the first version of this work. The new studied location supports the previously presented data and, although it is located also in a brackish environment, the authors added enough evidence to suppress my doubts regarding the influence of salinity in methanogenesis.

Regarding my first specific comment from my previous report ("for the reasons previously stated, I think that these ecosystems cannot be described as globally prevalent"), I was referring to the brackish water characteristic, topic that the authors have conveniently dealt with in this version of the manuscript. I completely agree with the authors about the global importance of macroalgal habitats.

I also agree with the recommendations of reviewer #2 about moving the focus of the manuscript away from blue carbon and I think that following them have improved the overall quality of this work.

Specific comments

I did not find the environmental variables for the new location in the rocky island in the Finnish archipelago, neither in the manuscript nor in the supplementary material; they could be added to the supplementary material.

Regarding the new literature about methane emissions in saline and hypersaline environments, I did not find the citation for Beversdorf et al. 2008 in the revised manuscript.

Figure S7 A and S7 B both represent the same data and one of them and the authors could remove one of them to avoid redundancy.

Reviewer #2 (Remarks to the Author):

The manuscript by Roth et al. has substantially improved from the first version and I appreciate how the authors have toned down the blue carbon discussion. However, there are still some concerns:

1)The three coastal ecosystems that are studied here are macroalgae, mixed vegetation, and bare sediments. Yet the title suggests that this paper is about coastal vegetation (macroalgae and mixed vegetation) and simply ignores bare sediments. But the GHG dynamics in bare sediments are a major part of this study because they have the second highest mean CH₄ fluxes (> than macroalgae) and e.g. higher CO₂ fluxes in the winter than macroalgae. However, most of the discussion is around vegetated coastal ecosystems. The abstract starts with an introduction about the importance of vegetated coastal ecosystems. Indeed, this has been the focus of blue carbon assessments, but your study shows that bare sediments are equally important. In fact, I believe the greatest CO₂ offsets by CH₄ were found in bare sediments in fall. Why is the focus then on the vegetated ecosystems here in the offset discussion? I would suggest shifting this and discuss these high offsets in bare sediments. What do we learn from this? Does this have any new management implications?

2)While you now avoid the term "blue carbon" you still compare your CH₄ fluxes to the "carbon sink capacity", which I believe is still misleading. I suggest you have two options here: either clearly define carbon sink capacity and make sure the reader understands that you only refer to the water-air CO₂ uptake here or avoid this term altogether and simply use CO₂ flux, which is what you measured. You can only make certain assumptions about the ecosystem carbon sink capacity but have no evidence how much of CO₂ is buried or exported. This is especially important

because the studied coastal ecosystems here have very fluent boundaries to other marine habitats.

Other minor comments:

L24: I would rewrite this. CH₄ emissions and offsets in coastal vegetated ecosystems have been studied in detail in previous studies (e.g. Al-Haj and Fulweiler 2020, Oreska et al. 2020, Rosentreter et al. 2018 etc...). Maybe try to highlight more the novelty that these are the first CH₄ fluxes measured from macroalgae (is this correct?) and the importance of vegetated and unvegetated coastal ecosystems.

L27 I would delete "all". It reads like all habitats have this exact range, but I believe you mean overall and across all ecosystem types you found a range of 0.1-2.9 mg CH₄ m⁻² d⁻¹.

L45 delete be

L76-78: Is this correct? I believe most studies measure CO₂ and CH₄ simultaneously and separately from N₂O. Especially since many research groups use Picarro these days. This method has been used in several studies before and therefore is not very novel per se.

L81 It is a bit unclear to me what you are trying to state here. CO₂ and CH₄ are often measured simultaneously.

L119-121: please include the statistics. I would suggest to always present mean (or median) and an uncertainty (either Stdeva, SE, c.i. or a range) with each number you report.

L122 Here you mention the large spatiotemporal heterogeneity but in the sentence before the means that you present (0.34, 0.38 and 0.55) are fairly close and not highly variable. I would suggest to re-phrase this. Another reason to please include uncertainties.

Generally, I would discuss the CH₄ fluxes between the three habitats a bit more here. Are you not surprised that on average bare sediments have higher CH₄ fluxes than macroalgae? Can you provide a few suggestions why you think this is the case?

In the next paragraph please also present the mean (or median) and uncertainty of CO₂ fluxes for each of the three habitats over the annual cycle. I was unable to find this important information in the main ms.

L163 When I look at Fig 2A and B, I don't find the CO₂ offset by CH₄ emissions substantial. Only in the fall in macroalgae, mixed vegetation, and bare sediments the offset seems to be substantial but in all other seasons the offsets seem very small. What is the overall annual offset in each of the three habitats? Again, here you discuss mostly vegetated habitats while I believe you find the largest offsets in bare sediments?

L301-302 do you mean Fig 2A?

L306-307 Yes if you are the first to ever report CH₄ emission from macroalage, I think this should be highlighted somewhere earlier. Are the authors aware of the study of marine macroalgae in vitro CH₄ production (Machado L, Magnusson M, Paul NA, de Nys R, Tomkins N (2014) Effects of Marine and Freshwater Macroalgae on In Vitro Total Gas and Methane Production. PLoS ONE 9(1): e85289. And Hansson, G. "Methane production from marine, green macro-algae." Resources and conservation 8.3 (1983): 185-194. Maybe if these studies are somewhat comparable to yours it would be interesting to see how the production rates compare?

Reviewer #3 (Remarks to the Author):

I consider that the authors have adequately responded to the revisions made. Not only that, they have also improved the work by adding new data; the sequencing of the 16S rRNA gene of floating

algae and/or organic matter debris associated with macroalgae, as well as extending the sampling to the mediations. All of this together provides a more robust basis for their conclusions. Therefore, I consider that this work is suitable for publication in the journal.

Reviewer #1 (Remarks to the Author):

After reviewing the revised manuscript, I find that it has improved significantly, with new data and references regarding the major concerns that I had with the first version of this work. The new studied location supports the previously presented data and, although it is located also in a brackish environment, the authors added enough evidence to suppress my doubts regarding the influence of salinity in methanogenesis. Regarding my first specific comment from my previous report (“for the reasons previously stated, I think that these ecosystems cannot be described as globally prevalent”), I was referring to the brackish water characteristic, topic that the authors have conveniently dealt with in this version of the manuscript. I completely agree with the authors about the global importance of macroalgal habitats. I also agree with the recommendations of reviewer #2 about moving the focus of the manuscript away from blue carbon and I think that following them have improved the overall quality of this work.

Response: Thank you very much for the positive feedback. Your previous suggestions have been very valuable to improving the manuscript. Thank you also for re-reading the manuscript and the below specific comments. We have addressed them in a final revision.

Specific comments

I did not find the environmental variables for the new location in the rocky island in the Finnish archipelago, neither in the manuscript nor in the supplementary material; they could be added to the supplementary material.

Response: We added the environmental conditions from the Finnish archipelago to the figure description of Supplementary Figure 8.

“The concentration was recorded for approximately 1h continuously at each location between 09:00 – 12:00 on October 13, 2021. During this period, the following environmental conditions were recorded across the three stations (mean \pm SD): water temperature = 11.34 ± 0.03 °C; salinity = 5.98 ± 0.00 ; windspeed = 3.4 ± 0.6 m/s.”

Regarding the new literature about methane emissions in saline and hypersaline environments, I did not find the citation for Beversdorf et al. 2008 in the revised manuscript.

Response: We added the reference to the manuscript. However, there has also been a citation error from our side: The intended study is: *Karl DM, Beversdorf L, Björkman KM, Church MJ, Martinez A, Delong EF. Aerobic production of methane in the sea. Nat Geosci. 2008;1:473–8.*

We rephrased our original statement in the manuscript to the following:

“Importantly, methylotrophic methanogenesis can proceed in saline to hypersaline environments with high ambient sulfate concentrations^{50,60,61} and is, thus, expected to also play an important role in coastal environments with higher salinity compared to the brackish waters of the Baltic Sea.” Lines 278 – 281

Figure S7 A and S7 B both represent the same data and one of them and the authors could remove one of them to avoid redundancy.

Response: We have removed the panel Supplementary Fig. 7a from the Supplementary Material. In the revised version of the manuscript, the original Supplementary Fig. 7b is now referred to as Supplementary Fig. 7.

Reviewer #2 (Remarks to the Author):

The manuscript by Roth et al. has substantially improved from the first version and I appreciate how the authors have toned down the blue carbon discussion. However, there are still some concerns:

1) The three coastal ecosystems that are studied here are macroalgae, mixed vegetation, and bare sediments. Yet the title suggests that this paper is about coastal vegetation (macroalgae and mixed vegetation) and simply ignores bare sediments. But the GHG dynamics in bare sediments are a major part of this study because they have the second highest mean CH₄ fluxes (> than macroalgae) and e.g. higher CO₂ fluxes in the winter than macroalgae. However, most of the discussion is around vegetated coastal ecosystems. The abstract starts with an introduction about the importance of vegetated coastal ecosystems. Indeed, this has been the focus of blue carbon assessments, but your study shows that bare sediments are equally important. In fact, I believe the greatest CO₂ offsets by CH₄ were found in bare sediments in fall. Why is the focus then on the vegetated ecosystems here in the offset discussion? I would suggest shifting this and discuss these high offsets in bare sediments. What do we learn from this? Does this have any new management implications?

Response: Thank you for the positive feedback and this important remark. In the revised version of the manuscript, we have extended the discussion on why CH₄ and CO₂ fluxes differ between the habitat-types. We implemented this change at various instances throughout the manuscript, starting with title and abstract (that now include more details on the bare sediments), followed by detailed discussion in the results/discussion sections. Particularly the bare sediment habitat gained more focus; however, we kept new text concise to not distract from the key finding of our study – the in situ CH₄ emissions from macroalgae habitats.

Please refer to our responses to your specific comments below for all changes in the manuscript.

With regards to management implications: Over an annual cycle, the studied vegetated systems remain net sinks of CO₂-eq. fluxes, despite the concurrent CH₄ emissions. This contrasts bare sediment areas, which are net sources of carbon-based GHGs. As a result, the most obvious management implication is to protect vegetated systems from deterioration (and potential shift to sediment plains) by anthropogenic impacts. However, how changes in vegetation cover and biodiversity translate into functioning remains unresolved, for which we prefer to refrain from too speculative management implications. Nonetheless, in the revised version of the manuscript, we added the following text to the last section of the discussion:

“In conclusion, our simultaneous high-resolution sea-air CO₂ and CH₄ flux measurements show that CH₄ emissions can offset one-third of the carbon sink capacity attributed to atmospheric CO₂ uptake over an annual cycle across highly productive macroalgae and mixed vegetation coastal ecosystems in a northern temperate region. Net atmospheric CO₂ uptake still outweigh CO₂-eq. CH₄ emissions – that is, these habitats exert a net cooling impact over centurial timescales. This net radiative forcing benefit contrasts surrounding unvegetated sediment areas, which act as net atmospheric CO₂ and CH₄ source. Thus, the conservation and restoration of vegetated coastal ecosystems is advocated because it may effectively remove CO₂ from the atmosphere and reduce the adverse effects of climate change.” Lines 341 – 348

2) While you now avoid the term “blue carbon” you still compare your CH₄ fluxes to the “carbon sink capacity”, which I believe is still misleading. I suggest you have two options here: either clearly define carbon sink capacity and make sure the reader understands that you only refer to the water-air CO₂ uptake here or avoid this term altogether and simply use CO₂ flux, which is what you measured. You can only make certain assumptions about the ecosystem carbon sink capacity but have no evidence how much of CO₂ is buried or exported. This is especially important because the studied coastal ecosystems here have very fluent boundaries to other marine habitats.

Response: To avoid potential for misinterpretation, we have now clearly defined the “carbon sink capacity attributed to atmospheric CO₂ uptake” in the text (lines 167 – 170)

“In the following, the carbon sink capacity attributed to atmospheric CO₂ uptake refers to an instantaneous influx of CO₂ from the atmosphere into the water caused by undersaturation of pCO₂ in surface waters relative to the atmospheric equilibrium; note, this capacity does not relate to long-term carbon sequestration processes as burial or export.”

Also, at each instance where we refer to this capacity, we specifically state “the carbon sink capacity attributed to atmospheric CO₂ uptake”, clearly distinguishing it from any carbon burial or other sedimentary processes.

Other minor comments:

L24: I would rewrite this. CH₄ emissions and offsets in coastal vegetated ecosystems have been studied in detail in previous studies (e.g. Al-Haj and Fulweiler 2020, Oreska et al. 2020, Rosentreter et al. 2018 etc...). Maybe try to highlight more the novelty that these are the first CH₄ fluxes measured from macroalgae (is this correct?) and the importance of vegetated and unvegetated coastal ecosystems.

Response: Thank you for the suggestion. In the revised and shortened abstract, we highlight better the knowledge gaps specific to our manuscript:

“Coastal ecosystems can efficiently remove carbon dioxide (CO₂) from the atmosphere and are thus promoted for nature-based climate change mitigation. Natural methane (CH₄) emissions from these ecosystems may counterbalance atmospheric CO₂ uptake. Still, knowledge of mechanisms sustaining CH₄ emissions from and the contribution to net radiative forcing remains scarce for globally prevalent macroalgae, mixed vegetation, and surrounding depositional sediment habitats.” Lines 22 – 26

L27 I would delete “all”. It reads like all habitats have this exact range, but I believe you mean overall and across all ecosystem types you found a range of 0.1-2.9 mg CH₄ m⁻² d⁻¹.

Response: Removed.

L45 delete be

Response: Removed.

L76-78: Is this correct? I believe most studies measure CO₂ and CH₄ simultaneously and separately from N₂O. Especially since many research groups use Picarro these days. This method has been used in several studies before and therefore is not very novel per se.

Response: Thank you for the constructive comment. You are right – simultaneous CO₂ and CH₄ measurements have been performed in coastal ecosystems before. They are, however, restricted to few assessments in mangrove and seagrass systems. In the revised version of the manuscript, we thus removed the claim about the lack of concurrent CO₂ and CH₄ measurements, but focus on how the application of in situ automated cavity ring-down spectroscopy has helped to improve our knowledge in mangrove and seagrass systems. In this context, we provide reference to the studies available and highlight how this technique should be expanded to other globally prevalent ecosystems, such as the here-studied macroalgae, mixed vegetation, and bare sediment habitats.

“Simultaneous and continuous CO₂ and CH₄ sea-air flux measurements are, therefore, indispensable to determine whether a coastal ecosystem acts as a net source or sink of atmospheric carbon-based GHGs – that is, if it has a positive or negative effect on radiative forcing²¹. In situ automated cavity ring-down spectroscopy is particularly effective to quantify coastal sea-air CO₂ and CH₄ fluxes simultaneously – but its application has been limited to estuarine, mangrove, and seagrass systems^{27–30}. The paucity of similar measurements across a wider range coastal environments – for example globally prevalent and highly productive macroalgae and mixed vegetation habitats, or their surrounding shallow depositional sediment areas – currently complicates efforts evaluating the realized potential of our coasts to remove carbon from the atmosphere. This is because 1) rigorous evidence for the uptake of atmospheric CO₂ by many coastal systems through direct sea-air CO₂ gas exchange remains understudied¹⁷, and 2) concurrent CH₄ emissions from these environments could offset or even negate their value as atmospheric carbon dioxide sinks⁷.” Lines 76 – 87

L81 It is a bit unclear to me what you are trying to state here. CO₂ and CH₄ are often measured simultaneously.

Response: Please see above comment.

L119-121: please include the statistics. I would suggest to always present mean (or median) and an uncertainty

(either Stdeva, SE, c.i. or a range) with each number you report.

Response: We added uncertainties to all data presented in the text.

L122 Here you mention the large spatiotemporal heterogeneity but in the sentence before the means that you present (0.34, 0.38 and 0.55) are fairly close and not highly variable. I would suggest to re-phrase this. Another reason to please include uncertainties.

Response: The cumulative flux represents a first order estimate of the annual flux, which we now present with a propagated standard error based on daily means (\pm SE) of each season:

“Over an annual cycle, the cumulative net fluxes of CH₄ to the atmosphere were 0.34 (\pm 0.01) g CH₄ m⁻² y⁻¹ in the macroalgae, 0.55 (\pm 0.03) g CH₄ m⁻² y⁻¹ in the mixed vegetation, and 0.38 (\pm 0.02) g CH₄ m⁻² y⁻¹ in the surrounding bare sediment areas (data presented as cumulative annual net flux and propagated error using daily means and the associated uncertainty).” Lines 121 – 125

Accordingly, the data suggests differences of 11 – 61% across habitat-types. The importance for acknowledging these differences has been discussed in a recent publication, which we refer to with reference “39” at several instances throughout the manuscript.

Generally, I would discuss the CH₄ fluxes between the three habitats a bit more here. Are you not surprised that on average bare sediments have higher CH₄ fluxes than macroalgae? Can you provide a few suggestions why you think this is the case?

Response: On the contrary: High CH₄ fluxes from the sediment areas are expected because of the sediment biogeochemistry that promotes CH₄ production in organic matter rich coastal soft sediments. We have extended the discussion on why CH₄ fluxes differ between the various systems at various instances throughout the manuscript. We kept this discussion concise to not lose focus on the key finding of our study – the in situ CH₄ emissions from macroalgae habitats.

“High rates of CH₄ emissions have been ascribed to habitats with mixed vegetation^{40,51} and surrounding depositional^{52,53} areas with organic matter-rich soft sediments. In general, these and similar coastal sediment systems account for the majority of total marine CH₄ emissions²⁰. Rapid organic matter and sediment accumulation rates, deep anoxic sediment layers, bottom water oxygen depletion, and shallow sulfate-methane transition zones acting as “CH₄-filter” can all contribute to increased CH₄ release rates from coastal sediments^{15,46,54}. The high rates of CH₄ emissions from the macroalgae habitats in our study are therefore intriguing because of the prevalence on rocky hard-bottom substrates and the absence of the above-mentioned “classical” sedimentary conditions that promote CH₄ formation. To examine ...”
Lines 215 – 222

In the next paragraph please also present the mean (or median) and uncertainty of CO₂ fluxes for each of the three habitats over the annual cycle. I was unable to find this important information in the main ms.

Response: In line with the CH₄ fluxes, we have added the uncertainty of cumulative annual CO₂ fluxes:

“Over an annual cycle, the macroalgae and mixed vegetation habitats acted as net sinks of atmospheric CO₂ with cumulative fluxes of -52 (\pm 5) and -71 (\pm 10) g CO₂ m⁻² y⁻¹, respectively (data presented as cumulative annual net flux and propagated error using daily means and the associated uncertainty). In comparison, the bare sediments were net sources of CO₂ to the atmosphere with 30 (\pm 6) g CO₂ m⁻² y⁻¹.”

L163 When I look at Fig 2A and B, I don't find the CO₂ offset by CH₄ emissions substantial. Only in the fall in macroalgae, mixed vegetation, and bare sediments the offset seems to be substantial but in all other seasons the offsets seem very small. What is the overall annual offset in each of the three habitats? Again, here you discuss mostly vegetated habitats while I believe you find the largest offsets in bare sediments?

Response: The CO₂ offset by CO₂-eq. CH₄ emissions covers a wide range of values across seasons and habitat-types. Indeed, the offset is minimal in winter, but reaches already 5, 12 and 29% in spring (macroalgae, mixed vegetation, bare sediments, respectively). Seasonally, the greatest offsets by CO₂-eq. CH₄ fluxes were observed in summer and fall, reducing the net GHG balance by 21 – 44% in the

macroalgae, 16 – 47% in the mixed vegetation, and 42 – 84% in the bare sediment habitats (please refer to lines 180 – 183).

We also report the annual offset in each of the three habitats in lines 187 – 191: “Over an annual cycle, CO₂-eq. CH₄ fluxes lowered the net atmospheric CO₂-eq. sink capacity attributed to CO₂ uptake in the macroalgae habitats from -52 down to -38 g CO₂-eq. m⁻² y⁻¹ (i.e., 28% reduction) and from -71 down to -46 g CO₂-eq. m⁻² y⁻¹ (i.e., 35% reduction) in the mixed vegetation habitats. Cumulative CO₂-eq. CH₄ fluxes augmented the positive net sea-air CO₂ fluxes of bare sediments by 57% (i.e., from 30 to 47 g CO₂-eq. m⁻² y⁻¹).”

Accordingly, if CO₂-eq. CH₄ emissions were not accounted for in an annual budget, the net atmospheric CO₂-eq. sink capacity attributed to CO₂ influxes may be overestimated by around 1/3 in each of the habitats. We removed “substantially” from the text to avoid any subjective judgement.

Regarding the large offset in bare sediments. It is an interesting result but also expected. As stated in lines 174 – 180 the offset depends on the magnitude of CO₂ relative to CO₂-eq. CH₄ fluxes. Sediments have little photosynthetic activity (e.g., through microphytobenthos) compared to vegetated systems. As a result, a low CO₂ influx can quickly be offset by concurrent CH₄ emissions. In the revised version, we added a short discussion on this matter in line 174 – 180:

“... however, the magnitude of this offset was variable across habitat types and seasons, depending on the magnitude of CO₂ relative to CO₂-eq. CH₄ fluxes (Fig. 2a). For example, some of the highest offset (i.e., 84%; Fig. 2, Supplementary Table 1) was observed in the bare sediment habitats, where photosynthetic activity by microphytobenthos or dislodged macrophytes (leading to CO₂ uptake) is counterbalanced by ecosystem respiration (leading to CO₂ release)³², and CH₄ fluxes are sustained by organic matter-rich soft sediments⁴⁶. In contrast, highly productive macroalgae (leading to increased rates of CO₂ uptake) and marginal CH₄ emissions showed generally lower offsets in the carbon sink capacity attributed to atmospheric CO₂ uptake by concurrent CH₄ emissions.”

L301-302 do you mean Fig 2A?

Response: Yes, we wanted to refer to Fig 2a and b. Thank you for pointing out.

L306-307 Yes if you are the first to ever report CH₄ emission from macroalage, I think this should be highlighted somewhere earlier. Are the authors aware of the study of marine macroalgae in vitro CH₄ production (Machado L, Magnusson M, Paul NA, de Nys R, Tomkins N (2014) Effects of Marine and Freshwater Macroalgae on In Vitro Total Gas and Methane Production. PLoS ONE 9(1): e85289. And Hansson, G. "Methane production from marine, green macro-algae." Resources and conservation 8.3 (1983): 185-194. Maybe if these studies are somewhat comparable to yours it would be interesting to see how the production rates compare?

Response: To the best of our knowledge, we are the first to report in situ CH₄ emissions from marine macroalgae. The suggested references refer to in vitro studies where macroalgae material is artificially fermented in a digester. A comparison of rates to such studies would not be very helpful. Although novelty claims (whether true or not) are not supported by Nature Communications, we have now highlighted in the results/discussion that only such in vitro studies exist so far. In addition, there is one new study reporting on the natural degradation of macroalgae (also *Fucus vesiculosus*) on beaches as beach wrack: <http://dx.doi.org/10.1007/s13280-022-01774-4>. The new text is found in lines 110 – 112:

“We report in situ CH₄ sea-air fluxes in the order of 0.1 ± 0.0 to 1.8 ± 0.1 mg CH₄ m⁻² d⁻¹ (mean ± SE) from the macroalgae habitat. CH₄ emissions from macroalgae have previously only been reported from in vitro studies where macroalgae material was artificially fermented^{35,36} or in situ from natural degradation of macroalgae (i.e., *Fucus vesiculosus*) on beaches as beach wrack³⁷.”

Reviewer #3 (Remarks to the Author):

I consider that the authors have adequately responded to the revisions made. Not only that, they have also improved the work by adding new data; the sequencing of the 16S rRNA gene of floating algae and/or organic matter debris associated with macroalgae, as well as extending the sampling to the mediations. All of this together provides a more robust basis for their conclusions. Therefore, I consider that this work is suitable for publication in the journal.

Response: Thank you very much for the positive response and the suggestions along the way, which have significantly contributed to improving the manuscript.